# COMPLEXITY- AND STATISTICS-GUIDED ANOMALY DETECTION IN TIME SERIES FOUNDATION MODELS

**Jongwon Kim**[1]     **Samuel Yoon**[1]     **Young Myoung Ko**[1]*     **Yerin Kim**[2]     **Sung Il Kim**[2]
**Jaeung Tae**[2]

[1]POSTECH, {pioneer0517, muel2, youngko}@postech.ac.kr
[2]SK hynix Inc., {yerin1.kim, sungil2.kim, jaeung.tae}@sk.com

## ABSTRACT

This paper introduces a methodology for anomaly detection in time series using Time Series Foundation Models (TFMs). While TFMs have achieved strong success in forecasting, their role in anomaly detection remains underexplored. We identify two key challenges when applying TFMs to reconstruction-based anomaly detection and propose solutions. The first challenge is overgeneralization, where TFMs reconstruct both normal and abnormal data with similar accuracy, masking true anomalies. We find that this effect often occurs in data with strong low-frequency components. To address it, we propose a complexity metric, $\alpha$, that reflects how difficult the data is for TFMs and design a Complexity-Aware Ensemble (CAE) that adaptively balances TFMs with a statistical model. The second challenge is overstationarization, caused by instance normalization layers that improve forecasting accuracy but remove essential statistical features such as mean and variance, which are critical for anomaly detection. We resolve this by reintroducing these features into the reconstruction process without retraining the TFMs. Experiments on 23 univariate and 17 multivariate benchmark datasets demonstrate that our method significantly outperforms both deep learning and statistical baselines. Furthermore, we show that our complexity-based metric, $\alpha$, provides a theoretical foundation for improved anomaly detection, and we briefly explore prediction-based anomaly detection using TFMs.

## KEYWORDS

Time series anomaly detection, Time series foundation model

## 1 INTRODUCTION

Anomaly detection identifies events or points that deviate from expected patterns and is context dependent (Chandola et al., 2009). In time series, it plays a vital role in industrial monitoring (Zheng et al., 2017), healthcare (Kiranyaz et al., 2015), safety systems (Giraldo et al., 2018), and finance (Ahmed et al., 2016). The emergence of foundation models, inspired by the success of large language models (LLMs), has opened new opportunities for time-series analysis. Recent advances in time-series foundation models (TFMs), whether LLM based or independently designed, have demonstrated strong predictive capabilities (Wu et al., 2023; Gao et al., 2024; Zhou et al., 2023; Woo et al., 2024; Goswami et al., 2024; Liang et al., 2024; Ansari et al., 2024; Shi et al., 2025; Liu et al., 2025). However, despite their superior performance on prediction tasks, applications of TFMs to anomaly detection remain limited.

This study addresses two primary limitations of TFMs for time-series anomaly detection in reconstruction settings. In reconstruction-based anomaly detection, anomalies are identified via reconstruction error, with larger errors indicating anomalous points. The first limitation, overgeneralization, occurs when a model reconstructs both normal and anomalous data with similarly high accuracy, yielding low anomaly scores even for truly anomalous cases. Prior work has largely attributed this behavior

---

*Corresponding author.

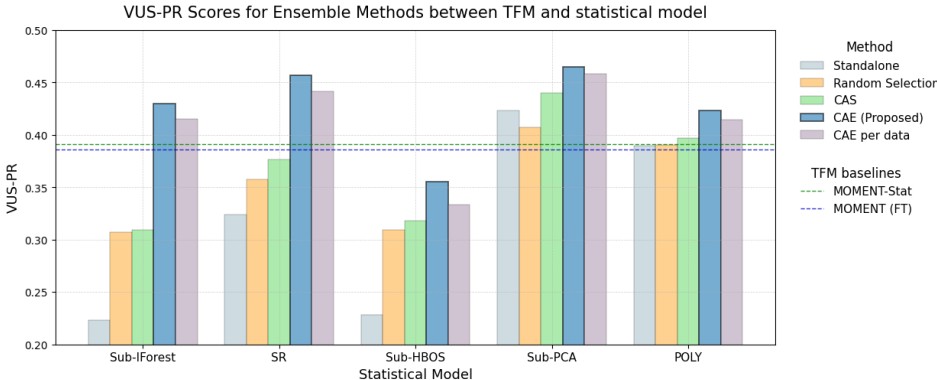

Figure 1: VUS-PR scores for five statistical models (Sub-IForest (Liu et al., 2008), SR (Ren et al., 2019), Sub-HBOS (Goldstein & Dengel, 2012), Sub-PCA (Aggarwal & Aggarwal, 2017), and POLY (Li et al., 2007)) combined with MOMENT, a time-series foundation model. Within each model group, bars indicate the standalone statistical method and four ensemble strategies: Random Selection, CAS, CAE (Proposed), and CAE per data (our model variants). The green and blue dashed lines represent the MOMENT-Stat and fine-tuned MOMENT (FT) baselines, respectively.

to high model capacity while paying less attention to the role of data complexity. We show that overgeneralization becomes more pronounced on simpler datasets, for example those dominated by low-frequency structure, where the capacity of TFMs enables easy reconstruction of anomalies (Gong et al., 2019; Park et al., 2020; Wang et al., 2022). The second limitation, overstationarization, arises when instance-normalization layers remove critical distributional characteristics such as mean and variance. Although instance normalization improves predictive performance (Kim et al., 2021; Liu et al., 2022), it can hinder anomaly detection.

To overcome these limitations, we propose two complementary approaches. First, we introduce a complexity-aware ensemble (CAE) that uses a complexity metric $\alpha$, derived from the TFM output, to adaptively balance TFMs with simpler statistical models based on dataset complexity. This mechanism prefers statistical methods for low-complexity data and TFMs for high-complexity data. We theoretically show that $\alpha$ is sensitive to high-frequency content and, under moderate assumptions, that larger $\alpha$ provides a provable guarantee of greater separation between normal and anomalous scores. Second, we improve reconstruction by explicitly reintroducing instance-level mean and variance into the encoder output, thereby mitigating overstationarization without retraining the TFM. Our experiments across 23 univariate benchmarks demonstrate the effectiveness of both approaches.

Figure 1 shows that CAE outperforms the MOMENT-Stat baseline, standalone models, and alternative ensemble strategies. Moreover, the overstationarization-corrected TFM (MOMENT-Stat) achieves higher anomaly-detection performance than fine-tuned MOMENT. By excluding the poorly performing Sub-HBOS variant, the CAE over the remaining four statistical models further surpasses the previous state-of-the-art Sub-PCA score of 0.4233 as well as the VUS-PR of 0.3857 achieved by the vanilla TFM, while our proposed method attains a score of 0.4679.

We also evaluate forecasting-oriented TFMs for anomaly detection by interpreting higher forecasting errors as anomalies. While forecasting models have traditionally underperformed due to limited predictive accuracy, recent TFMs motivate a reexamination of this paradigm. In addition, many modern TFM architectures, often inspired by PatchTST, do not explicitly model inter-channel dependencies despite strong multivariate forecasting performance. We evaluate these models on multivariate anomaly-detection tasks and show that capturing relationships among channels is critical for effective detection.

The remainder of the paper is organized as follows: Section 2 reviews related work, Section 3 details the proposed methodology, Section 4 presents experimental results, and Section 5 concludes.

## 2 RELATED WORK

Prior work has investigated a variety of methods to mitigate overgeneralization and overstationarization in time-series anomaly detection. Overgeneralization—where models reconstruct both normal and anomalous data with similarly high accuracy—has been addressed by several strategies. Notable approaches include limiting the expressive capacity of decoders (Wang et al., 2022; Song et al., 2023; Goswami et al., 2024), using memory-based models (Song et al., 2023; Liu et al., 2024), applying contrastive learning with synthetic anomalies (Audibert et al., 2020; Sun et al., 2024; Wang et al., 2023; Liu et al., 2024), and adding auxiliary scoring mechanisms beyond plain reconstruction error (Su et al., 2019; Zhao et al., 2020; Tuli et al., 2022; Xu et al., 2022; Song et al., 2023).

Each strategy has limitations. Reducing decoder capacity does not eliminate overgeneralization, even when the decoder is a single fully connected layer (Goswami et al., 2024). Memory-based methods typically require retraining and can be computationally expensive for large TFMs. Contrastive schemes struggle to produce meaningful synthetic anomalies because definitions of anomalies are dataset dependent; generic perturbations or noise may not reflect true anomalies. Finally, introducing auxiliary scores increases sensitivity to the choice and weighting of components. In contrast, we integrate a statistical-model score and adaptively set the weights from data characteristics, which reduces manual tuning.

Overstationarization arises when instance-normalization or RMSNorm layers remove essential statistics, such as mean and variance, from each input segment. These layers are common in TFMs (Zhou et al., 2023; Goswami et al., 2024; Woo et al., 2024; Liang et al., 2024; Ansari et al., 2024; Gao et al., 2024; Shi et al., 2025; Liu et al., 2025) because they improve predictive accuracy, yet they can hinder anomaly detection. Prior work (Kim et al., 2021; Liu et al., 2022) shows that re-injecting the lost statistics can improve forecasting. However, this idea has been less explored for anomaly detection, and naive implementations may require costly retraining. A practical alternative is to append per-instance statistics directly to token embeddings, which supplies distributional cues without additional training.

In this study, we use MOMENT (Goswami et al., 2024) as our reconstruction-based anomaly detection model, and Chronos (Ansari et al., 2024), Moirai (Woo et al., 2024), and TimeMoE (Shi et al., 2025) as forecasting-based models. MOMENT provides released parameters and explicit support for anomaly detection. Other TFMs that support anomaly detection, including One-fits-all (Zhou et al., 2023), TimesNet (Wu et al., 2023), and TimeMixer++ (Wang et al., 2025), do not release trained weights. MOMENT employs a Transformer architecture trained with masked imputation and has shown strong performance among TFMs (Liu & Paparrizos, 2024).

Forecasting-based anomaly detection has traditionally underperformed reconstruction-based methods because prediction errors on normal data can be substantial (Zamanzadeh Darban et al., 2024; Schmidl et al., 2022). Recent TFMs have improved forecasting quality; for example, PatchTST attains state-of-the-art results by treating each channel segment as an individual token while ignoring cross-channel relations (Nie et al., 2023). Whether higher forecasting accuracy translates into better anomaly detection, remains an open question. We address this gap through empirical evaluation.

A recent review (Zhang et al., 2024) surveys efforts to apply LLMs to time series, while subsequent studies report limited gains in forecasting (Tan et al., 2024) and anomaly detection (Zhou & Yu, 2025). We therefore focus on time-series foundation models rather than general-purpose LLMs.

## 3 METHOD

### 3.1 ANOMALY DETECTION USING A TIME-SERIES FOUNDATION MODEL

We consider a time series $x \in \mathbb{R}^T$ with instance normalization such that $\|x\|_2^2 = T$. We employ a pre-trained Time Series Foundation Model (TFM) consisting of an encoder $E : \mathbb{R}^T \to \mathbb{R}^d$ and a linear decoder $D : \mathbb{R}^d \to \mathbb{R}^T$. Given an input $x$ and a binary mask $M \in \{0, 1\}^T$ (where 0 indicates masked), the model output is $\hat{x}(M) = D(E(x \odot M))$.

In the standard reconstruction-based anomaly detection setting, we utilize two types of errors:

**Reconstruction Error ($\mathcal{L}_{\text{rec}}$):** The error computed when the model reconstructs the input with minimal or no masking (test-time configuration). Let $M_{\text{test}}$ be the test mask (often all ones or boundary masked).

$$\mathcal{L}_{\text{rec}}(x) = \|x - \hat{x}(M_{\text{test}})\|_2^2.$$

**Imputation Error ($\mathcal{L}_{\text{imp}}$):** The expected error when the model reconstructs the input under a random masking scheme used during pre-training (e.g., 30% masking). Let $\mathcal{M}$ be the distribution of random masks.

$$\mathcal{L}_{\text{imp}}(x) = \mathbb{E}_{M \sim \mathcal{M}} \left[ \|x - \hat{x}(M)\|_2^2 \right].$$

The anomaly score for a test instance is typically defined based on $\mathcal{L}_{\text{rec}}(x)$.

## 3.2 Complexity-aware ensemble and theoretical guarantees

We propose a *Complexity-Aware Ensemble (CAE)* that dynamically balances the TFM-based score with a statistical model score. The core motivation is that TFMs tend to overgeneralize on low-complexity data (where $\mathcal{L}_{\text{imp}} \approx \mathcal{L}_{\text{rec}}$), masking anomalies.

**Complexity Metric $\alpha$.** We define the raw complexity score $w(x)$ as the normalized difference between the imputation error (hard task) and the reconstruction error (easy task):

$$w(x) = \frac{\mathcal{L}_{\text{imp}}(x) - \mathcal{L}_{\text{rec}}(x)}{\|x\|_2^2}. \tag{1}$$

To handle varying scales across datasets, we compute the final complexity metric $\alpha \in [0, 1]$ by applying a Quantile Transform to $w(x)$ over the dataset:

$$\alpha = \text{QuantileTransform}(w(x)).$$

A larger $\alpha$ implies that the model struggles significantly more with partial masking, indicating complex, unpredictable patterns. The final ensemble score is given by:

$$s^{\text{CAE}} = \alpha \cdot s^{\text{TFM}} + (1 - \alpha) \cdot s^{\text{stat}}.$$

We now provide theoretical justifications connecting $\alpha$ to the spectral properties of the data and the anomaly detection gap. Detailed proofs are provided in Appendix A.

**Theorem 1** ($\alpha$ reflects High-Frequency Energy). *Let $\phi(k)$ be the energy of the signal $x$ at Haar wavelet scale $k$ (where higher $k$ corresponds to higher frequencies). The raw complexity gap $\Delta = \mathcal{L}_{\text{imp}} - \mathcal{L}_{\text{rec}}$ decomposes as:*

$$\Delta \approx \sum_k \phi(k)\, b_k,$$

*where $b_k \geq 0$ is a scale-dependent coefficient. Under standard assumptions on TFM masking, $b_k$ is non-decreasing with respect to $k$.*

*Interpretation.* Since $b_k$ increases with $k$, the gap $\Delta$ (and thus $\alpha$) is a weighted sum of spectral energies where high-frequency components are weighted more heavily. Therefore, a higher $\alpha$ indicates that the time series contains significant high-frequency detail, which is harder for the TFM to interpolate from context.

**Theorem 2** (Gap Growth with High-Frequency Complexity). *Consider a gradient-based update of the decoder parameters $W$ to minimize the reconstruction loss on normal data $L_N(W)$. Let $L_A(W)$ be the loss on anomalous data, and let $t$ denote the training step index. Under the assumption that gradients for normal and anomalous data are not aligned, the change in the anomaly gap $\Delta_{\text{gap}} = L_A - L_N$ after a small step $\rho$ satisfies:*

$$\Delta_{\text{gap}}(W_{t+1}) - \Delta_{\text{gap}}(W_t) \geq \rho \, \|\nabla_W L_N(W_t)\|_F^2 + O(\rho^2).$$

*Furthermore, the gradient norm $\|\nabla_W L_N(W_t)\|_F^2$ is strictly increasing with the high-frequency information share of the normal data.*

*Interpretation.* Theorem 2 suggests that for high-complexity data (large $\alpha$), the model's optimization landscape is steeper ($\|\nabla L_N\|^2$ is large). This stems from the property that TFMs tend to preserve low-frequency components while struggling with high-frequency details (Assumption 2 in Appendix). Consequently, this leads to a larger separation ($\Delta_{\text{gap}}$) between normal and anomalous scores during the learning process.

### 3.3 PROPOSED ENHANCEMENT: INSTANCE STATISTICS AUGMENTATION

The second challenge, overstationarization, arises from instance normalization layers (e.g., RevIN, RMSNorm) inherent in TFMs. While these layers stabilize training, they discard first- and second-order statistics (mean $\mu$ and standard deviation $\sigma$). We formalize the limitation of this discarding process as follows:

**Lemma 3** (Affine Invariance under Instance Normalization). *Let $\mathcal{N}(x) = (x - \mu_x)/\sigma_x$ be the instance normalization, and let $f(x) = D(E(\mathcal{N}(x)))$ be the reconstruction model composed of an encoder $E$ and decoder $D$. For any input time series $x$ and its affine transformation $x' = \alpha x + \beta$ (with $\alpha > 0$), the normalized inputs are identical: $\mathcal{N}(x') = \mathcal{N}(x)$. Consequently,*

$$f(x') = D(E(\mathcal{N}(x'))) = D(E(\mathcal{N}(x))) = f(x).$$

This implies that the model $f$ is theoretically invariant to shifts in mean and variance, making it incapable of distinguishing statistical anomalies from normal data.

To resolve this, we propose **MOMENT-Stat**, which explicitly re-injects these statistics into the reconstruction process without retraining the massive encoder. Formally, let $F_i = E(\text{RevIN}(x_i))$ be the feature representation of the normalized input. We augment the decoding process by directly concatenating the instance statistics to the feature vector before the linear decoder $D$:

$$\hat{x}_i = \text{RevIN}^{-1}\left(D([F_i; \mu_i; \sigma_i]), \mu_i, \sigma_i\right).$$

This simple modification ensures that the reconstruction error $\mathcal{L}_{\text{rec}}$ is sensitive to abnormal shifts in $\mu$ and $\sigma$, significantly boosting performance on datasets where anomalies are statistical outliers rather than pattern violations.

## 4 EXPERIMENT

We evaluate our method on the univariate TSB-AD-U benchmark, which contains 23 datasets from diverse domains such as web services, stock trading, healthcare, and industrial systems. Several univariate series were extracted from multivariate sources by selecting informative channels based on correlation and anomaly-detection performance. Results are reported as averages across datasets, following benchmark procedures in (Liu & Paparrizos, 2024). Detailed dataset descriptions are provided in the appendix.

MOMENT was trained with learning rate $10^{-4}$ for 2 epochs using a window size of 256 and the Adam optimizer. Training data for each dataset consisted of the first 25% of the time span or data until the first anomaly. The reconstruction head was a single fully connected layer with SiLU activation and dropout rate 0.1.

### 4.1 EVALUATION METRIC

Our primary metric is the Volume Under the Precision–Recall Surface (VUS-PR), as used in recent benchmarks (Paparrizos et al., 2022; Liu & Paparrizos, 2024; Goswami et al., 2024; Yang et al., 2023). VUS-PR ranges from 0 to 1, where larger values indicate stronger detection performance. Unlike threshold-based scores such as adjusted $F_1$, which may fluctuate due to arbitrary cutoffs or misalignments, VUS-PR remains stable and threshold-free. We also report complementary threshold-independent metrics: AUC-PR, AUC-ROC, and VUS-ROC. Full metric details appear in the appendix.

## 4.2 STATISTICAL MODELING AND TIME COMPLEXITY

Among 25 baseline detectors (Liu & Paparrizos, 2024), we focus on statistical models with time complexity not exceeding $\mathcal{O}(N \log N)$. These include SR (Ren et al., 2019), Sub-PCA (Aggarwal & Aggarwal, 2017), Sub-HBOS (Goldstein & Dengel, 2012), Sub-IForest (Liu et al., 2008), and POLY (Li et al., 2007). Their asymptotic complexities are summarized in Table 1, where $N$ is the number of points, $d$ the number of channels, $t$ the number of trees, and $p$ the polynomial order. The runtime of CAE equals the sum of the individual statistical models plus the cost of computing $\alpha$. MOMENT-Stat adds negligible runtime compared to fine-tuned MOMENT. Empirical timings are given in the appendix.

Table 1: Time complexity of selected models.

| Model | Time Complexity |
|---|---|
| SR | $\mathcal{O}(N \log N)$ |
| Sub-PCA | $\mathcal{O}(Nd^2 + d^3)$ |
| Sub-HBOS | $\mathcal{O}(Nd)$ |
| Sub-IForest | $\mathcal{O}(tN \log N)$ |
| POLY | $\mathcal{O}(Np^2 + p^3)$ |

## 4.3 OVERGENERALIZATION IN TFM RECONSTRUCTION

Figure 2 illustrates MOMENT's tendency to overgeneralize by reconstructing normal and anomalous data similarly, especially on lower-complexity datasets. We analyze three synthetic datasets—linear trends, quadratic functions, and Brownian motions—whose average complexities $\bar{\alpha}$ are 0.0660, 0.1034, and 0.4053, respectively. While PCA shows clearer gaps on low-complexity series but struggles on complex signals, MOMENT sometimes fails to separate anomalies in simple datasets. CAE mitigates both issues by combining MOMENT-Stat with a statistical model according to dataset complexity.

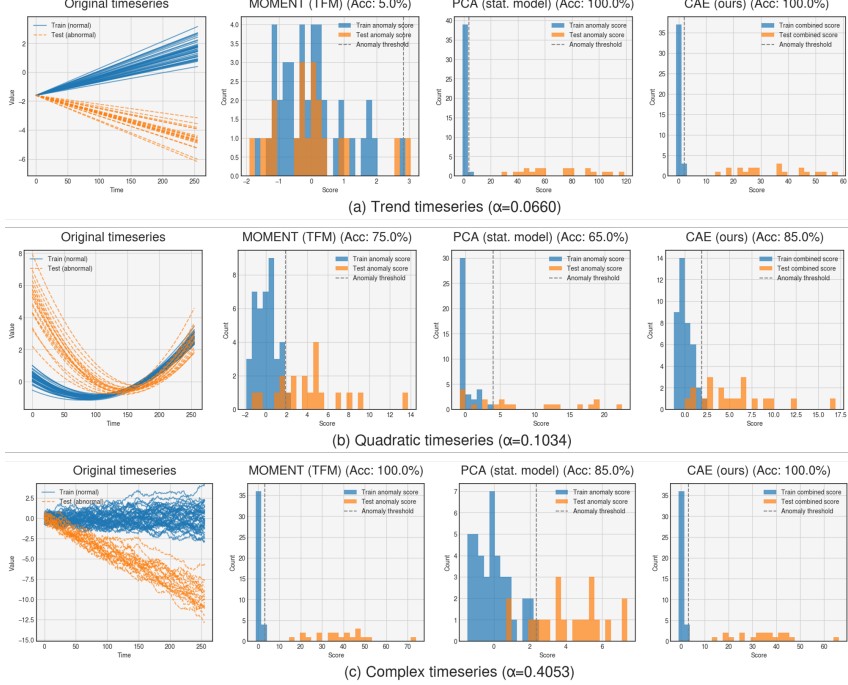

Figure 2: Three datasets are shown: (a) linear trends, (b) quadratic curves, and (c) complex Brownian-style signals with complexity measure $\alpha$. In each row, the leftmost panel displays the original training and anomalous test series. The second panels compare histograms of TFM reconstruction-based anomaly scores (dashed line = decision threshold). The next panels show PCA-based scores. Finally, our CAE ensemble clearly separates normal and anomalous samples.

## 4.4 PERFORMANCE OF INSTANCE STATISTICS AUGMENTATION

To address overstationarization, MOMENT-Stat incorporates mean and variance into the reconstruction phase. Table 2 compares MOMENT-Stat with a fine-tuned MOMENT model (MOMENT (FT)) and a zero-shot variant (MOMENT (ZS)) using a range of threshold-free metrics. Ranks are computed across 25 baselines in the benchmark suite. We include the global rank for reference even if a subset of methods is displayed in the table.

MOMENT-Stat attains a VUS-PR of 0.3913, outperforming MOMENT (FT) (0.3857) and MOMENT (ZS) (0.3790). This result suggests that reintroducing key statistical attributes (mean and variance) improves anomaly detection.

Table 2: Performance metrics comparison.

| Method | AUC-PR | AUC-ROC | VUS-PR | VUS-ROC | VUS-PR Rank |
|---|---|---|---|---|---|
| Sub-PCA | **0.3700** | 0.7100 | **0.4233** | 0.7600 | 1 |
| MOMENT-Stat | 0.3040 | **0.7146** | 0.3913 | **0.7771** | 3 |
| MOMENT (FT) | 0.3000 | 0.6900 | 0.3857 | 0.7600 | 6 |
| MOMENT (ZS) | 0.3000 | 0.6800 | 0.3790 | 0.7500 | 7 |

## 4.5 PERFORMANCE EVALUATION OF COMPLEXITY-AWARE ENSEMBLE

CAE fuses MOMENT-Stat with a statistical model to address overgeneralization. Table 3 shows VUS-PR scores for several models under different values of $\alpha$. An $\alpha = 0$ means only the statistical model is used, and $\alpha = 1$ means only the MOMENT-based model is used. Intermediate $\alpha$ values form various weighted combinations.

CAE improves VUS-PR for most statistical backbones. An exception occurs with Sub-HBOS, where the MOMENT-only setting ($\alpha = 1$) remains stronger than CAE. This is because the standalone performance of Sub-HBOS is substantially lower than that of MOMENT-Stat (0.2283 vs. 0.3913), so the ensemble gains little from the statistical component; in such cases, relying on a single strong model is preferable to blending with a weak one. Nevertheless, CAE still improves markedly over the standalone Sub-HBOS (0.3734 vs. 0.2283, rank 28→3). For example, Sub-PCA alone achieves a VUS-PR of 0.4233—the highest among existing baselines—which increases to 0.4679 when integrated within the CAE framework.

Table 3: VUS-PR scores for various models under different $\alpha$ settings, along with their rank changes relative to the 25 baselines in Table 11. Rank change indicates the position at $\alpha = 0$ and the new position after applying CAE.

| Model | $\alpha = 0$ | $\alpha = 0.1$ | $\alpha = 0.3$ | $\alpha = 0.5$ | $\alpha = 0.7$ | $\alpha = 1$ | CAE | Rank change |
|---|---|---|---|---|---|---|---|---|
| Sub-IForest | 0.2230 | 0.3685 | 0.3903 | 0.4155 | 0.4194 | 0.3913 | **0.4318** | 29 → 2 |
| SR | 0.3237 | 0.4006 | 0.4342 | 0.4376 | 0.4129 | 0.3913 | **0.4596** | 14 → 1 |
| Sub-HBOS | 0.2283 | 0.2522 | 0.3126 | 0.3688 | 0.3812 | **0.3913** | 0.3734 | 28 → 3 |
| Sub-PCA | 0.4233 | 0.4405 | 0.4560 | 0.4557 | 0.4448 | 0.3913 | **0.4679** | 1 → 1 |
| POLY | 0.3897 | 0.4004 | 0.4139 | 0.4170 | 0.4127 | 0.3913 | **0.4274** | 3 → 2 |

Table 3 shows that CAE consistently raises baseline models to the top of the benchmark. For example, SR improves from 14th to 1st, and Sub-IForest from 29th to 2nd. Sub-PCA remains 1st, while POLY and Sub-HBOS both move into the top three. Overall, CAE markedly boosts the ranking of statistical backbones.

## 4.6 COMPARISON WITH OTHER COMPLEXITY MEASURES

We also tested the other time-series complexity measures: Spectral Entropy (Inouye et al., 1991), Approximate Entropy (Pincus, 1991), and Sample Entropy (Richman & Moorman, 2000) for CAE. Table 4 reports the VUS-PR values for CAE under different complexity measures. For instance, the Sub-PCA model achieves a VUS-PR value of 0.4679 when using the proposed complexity

measure, which is higher than the VUS-PR values obtained when using the other complexity metrics. This indicates that our proposed measure leads to better anomaly detection performance across the evaluated models.

Table 4: VUS-PR Values for CAE with MOMENT-Stat under different complexity measures.

| Model Name | App Entropy | Sample Entropy | Spectral Entropy | Proposed Complexity |
|---|---|---|---|---|
| Sub-IForest | 0.3896 | 0.3928 | 0.3912 | **0.4318** |
| SR | 0.3928 | 0.3921 | 0.4515 | **0.4596** |
| Sub-HBOS | 0.3554 | 0.3548 | **0.3835** | 0.3734 |
| Sub-PCA | 0.4373 | 0.4409 | 0.4288 | **0.4679** |
| POLY | 0.3993 | 0.3987 | 0.4153 | **0.4274** |

## 4.7 EVALUATION OF COMPLEXITY-AWARE SELECTION

Model selection based on dataset complexity was also examined as an extreme case of ensemble weighting, where a single model receives full weight. Specifically, MOMENT-Stat was chosen when the dataset-level complexity $\bar{\alpha}_i$ exceeded the median complexity of that dataset; otherwise, a statistical model was used. As a baseline, models were also selected at random with equal probability. In addition, we report a variant denoted as *CAE per data*, which replaces the dataset-level complexity $\bar{\alpha}_i$ with the instance-level complexity $\alpha_i$. Table 5 shows that CAS improves VUS-PR scores compared to both the standalone and random-selection approaches. For example, Sub-PCA under CAS achieves a VUS-PR of 0.4400, outperforming vanilla Sub-PCA (0.4233) and random selection (0.4073). However, although CAS yields notable gains over simpler methods, CAE consistently achieves higher VUS-PR scores than CAS.

Table 5: Combined VUS-PR scores for merged models.

| Model | Vanilla | Random Selection | CAS | CAE | CAE per data |
|---|---|---|---|---|---|
| Sub-IForest | 0.2230 | 0.3071 | 0.3092 | **0.4318** | 0.4156 |
| SR | 0.3237 | 0.3575 | 0.3764 | **0.4596** | 0.4414 |
| Sub-HBOS | 0.2283 | 0.3097 | 0.3182 | **0.3734** | 0.3331 |
| Sub-PCA | 0.4233 | 0.4073 | 0.4400 | **0.4679** | 0.4582 |
| POLY | 0.3897 | 0.3905 | 0.3973 | **0.4274** | 0.4146 |

## 4.8 FORECASTING-ORIENTED TFM FOR ANOMALY DETECTION

Many prior studies have observed that reconstruction-based anomaly detectors generally outperform forecasting-based models, primarily because limited prediction accuracy results in high errors even for normal sequences. However, the emergence of large-scale TFMs has significantly enhanced forecasting capabilities.

Table 6 compares forecasting-oriented TFMs—Chronos (Ansari et al., 2024), TimeMoE (Shi et al., 2025), and Moirai (Woo et al., 2024)—under a zero-shot anomaly detection setting. Each model's anomaly detection performance metrics (AUC-PR, AUC-ROC, VUS-PR, VUS-ROC) and forecasting error (sMAPE) were evaluated individually. Here, the sMAPE is defined as $\text{sMAPE}(x, \hat{x}) = \frac{1}{T} \sum_{t=1}^{T} \frac{2|x_t - \hat{x}_t|}{|x_t| + |\hat{x}_t|}$. Chronos variants consistently achieve the best performance, followed closely by TimeMoE. Moirai, despite higher forecasting errors, shows notably lower anomaly detection performance.

Additionally, Figure 3 visually illustrates the relationship between forecasting accuracy (mean sMAPE) and anomaly detection performance (mean VUS-PR) for each model variant. A clear inverse relationship between forecasting error and anomaly detection performance is observed, suggesting that models with higher forecasting accuracy (lower sMAPE) generally yield better anomaly detection performance. This supports the hypothesis that improvements in predictive accuracy directly contribute to enhanced forecasting-based anomaly detection.

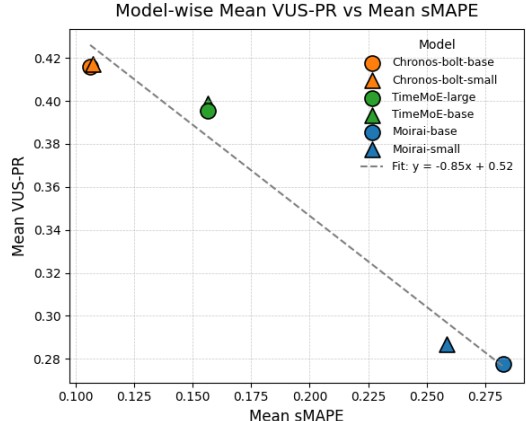

Figure 3: Relationship between forecasting accuracy (sMAPE) and anomaly detection performance (VUS-PR) for forecasting-oriented TFMs.

Table 6: Forecasting-based anomaly detection performance under zero-shot setting.

| Model | AUC-PR | AUC-ROC | VUS-PR | VUS-ROC | sMAPE |
|---|---|---|---|---|---|
| Chronos-bolt-base | 0.3901 | 0.7416 | 0.4160 | 0.8131 | 0.1063 |
| Chronos-bolt-small | 0.4057 | 0.7478 | 0.4172 | 0.8169 | 0.1073 |
| TimeMoE-large | 0.3725 | 0.7224 | 0.3954 | 0.7980 | 0.1562 |
| TimeMoE-base | 0.3766 | 0.7236 | 0.3986 | 0.7971 | 0.1560 |
| Moirai-base | 0.2361 | 0.6914 | 0.2775 | 0.7865 | 0.2826 |
| Moirai-small | 0.2471 | 0.7056 | 0.2869 | 0.7916 | 0.2584 |

## 4.9 PERFORMANCE ON MULTIVARIATE TIME SERIES

We evaluated our framework on six multivariate datasets (MSL, SMD, SMAP, SWaT, Genesis, TAO). A limitation of MOMENT is its *channel-independence* assumption, which treats multivariate series as independent univariate channels, ignoring inter-channel dependencies. Consequently, simply averaging TFM and statistical scores (Naive Ensemble, $\alpha = 0.5$) often degrades performance compared to using Statistical Models alone ($\alpha = 0$), as shown in Table 7 (e.g., PCA VUS-PR drops from 0.3878 to 0.3132).

In contrast, our Complexity-Aware Ensemble (CAE) consistently achieves higher VUS-PR than the naive approach, validating the effectiveness of adaptive weighting. By adaptively regulating the TFM contribution based on data complexity, CAE effectively mitigates the drawbacks of channel-independence. Notably, CAE significantly improves VUS-PR for HBOS ($0.1751 \rightarrow 0.2535$) and LOF ($0.1091 \rightarrow 0.2122$), demonstrating that an intelligent weighting strategy is crucial for robust multivariate detection.

In contrast to the reconstruction experiments focused on a representative subset, Table 8 reports zero-shot anomaly detection results across the entire TSB-AD-M benchmark (17 datasets) for forecasting-based TFMs (Moirai, TimeMoE, Chronos) alongside reconstruction-based MOMENT. Among these models, only Moirai explicitly incorporates multi-channel information. Although Moirai uses a flattened multivariate sequence format similar to PatchTST, it uniquely constructs attention matrices considering channel-specific information. Notably, Moirai shows relatively low performance on univariate datasets but demonstrates significantly improved performance in multivariate scenarios. These results highlight the necessity of explicitly modeling inter-channel relationships in multivariate time-series anomaly detection, a requirement less critical in forecasting tasks. Rankings are computed based on VUS-PR, where lower values indicate superior performance relative to other baseline models.

Table 7: Performance comparison on multivariate datasets.

| Detector | VUS-PR | | | VUS-ROC | | |
|---|---|---|---|---|---|---|
| | Stat | Naive ($\alpha = 0.5$) | CAE | Stat | Naive ($\alpha = 0.5$) | CAE |
| HBOS | 0.1869 | 0.1751 | **0.2535** | 0.7148 | 0.7099 | **0.7324** |
| IForest | **0.3320** | 0.2553 | 0.2996 | **0.8139** | 0.8070 | 0.7981 |
| LOF | 0.1005 | 0.1091 | **0.2122** | 0.6300 | 0.6389 | **0.6796** |
| PCA | **0.3878** | 0.3132 | 0.3433 | **0.8026** | 0.7834 | 0.7730 |
| MOMENT | | 0.1887 | | | 0.6352 | |

Table 8: Multivariate anomaly-detection results under zero-shot setting.

| Model | AUC-PR | AUC-ROC | VUS-PR | VUS-ROC | Multi-channel | Rank |
|---|---|---|---|---|---|---|
| Chronos-bolt-base | 0.1605 | 0.5675 | 0.1672 | 0.6149 | False | 22 |
| TimeMoE-base | **0.2164** | 0.6074 | **0.2276** | 0.6611 | False | 13 |
| Moirai-base | 0.2023 | **0.6210** | 0.2142 | **0.6731** | True | 14 |
| MOMENT | 0.1427 | 0.5553 | 0.1989 | 0.6369 | False | 17 |

## 5 CONCLUSION

This study addresses two significant challenges in reconstruction-based TFM anomaly detection: overgeneralization and overstationarization. To mitigate these limitations, we proposed two main enhancements. First, we introduced a Complexity-Aware Ensemble (CAE) framework that adaptively integrates MOMENT-Stat with a statistical model based on dataset complexity. Second, Instance Statistics Augmentation preserves critical distributional information (mean and variance) during reconstruction, forming the MOMENT-Stat model. Importantly, this augmentation is easily adaptable to other TFMs without the need for retraining. Experimental evaluations on 23 univariate and 17 multivariate benchmark datasets confirmed that both MOMENT-Stat and CAE achieve state-of-the-art performance in anomaly detection, significantly outperforming existing baseline models.

Furthermore, we evaluated forecasting-based TFMs (Chronos, Moirai, TimeMoE) and found that forecasting accuracy correlates with anomaly-detection performance, with recent high-accuracy models achieving competitive, state-of-the-art results. In multivariate settings, only models that explicitly capture inter-channel dependencies (such as Moirai) perform effectively, highlighting the importance of cross-channel modeling.

One limitation of our work is that, when selecting the simple statistical model for the CAE framework, we arbitrarily restricted our choices to methods with $\mathcal{O}(N \log N)$ time complexity. Another limitation is our exclusive focus on reconstruction-based anomaly detection for univariate time series. Nonetheless, our concise yet meaningful experiments on forecasting-based detection and multivariate series suggest that TFMs can be extended to more general anomaly detection settings.

## 6 ACKNOWLEDGMENTS

This research was supported in part by SK Hynix AICC (P24.01); in part by the Korea Institute for Advancement of Technology (KIAT) grant funded by the Korea government (MOTIE) (No. RS-2024-00409092, 2024 HRD Program for Industrial Innovation); in part by the Korea Planning & Evaluation Institute of Industrial Technology (KEIT) grant funded by the Ministry of Trade, Industry and Energy (No. RS-2025-25458052, Development of Core Technologies for Manufacturing Foundation Models); in part by the National Research Foundation of Korea (NRF) grant funded by the Korea government (MSIT) (No. RS-2025-16072964); and in part by the Institute of Information & Communications Technology Planning & Evaluation (IITP) grant funded by the Korea government (MSIT) (No. IITP-2025-RS-2024-00441244, Global Data-X Leader HRD Program).

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

## APPENDIX A. PROOFS AND TECHNICAL DETAILS

This appendix provides the formal definitions, rigorous setups, and proofs for the theorems presented in Section 3. We establish the connection between the complexity metric $\alpha$ and the high-frequency spectral properties of the time series (Theorem 1) and provide the guarantee for the anomaly gap growth (Theorem 2).

APPENDIX A.1. COMMON SETUP AND NOTATION

**Signal, mask, and projections.** Let $x = (x_1, \ldots, x_T) \in \mathbb{R}^T$ be a zero-mean time series with instance normalization $\mathbb{E}\|x\|^2 = T$. Fix a masking ratio $p \in (0, 1)$ and an interval mask $I \subset \{1, \ldots, T\}$ with $|I| = \lfloor pT \rfloor$. Define the coordinate projections

$$(P_I x)_t = \mathbf{1}\{t \in I\} \, x_t, \qquad P_{I^c} = \mathrm{Id} - P_I.$$

**Haar wavelet expansion and energies.** On $[0, 1]$, let $\{\psi_{k,j}\}$ be the compactly supported Haar ONB (Daubechies, 1992). Identify $t \in \{1, \ldots, T\}$ with $t/T \in [0, 1]$ and expand

$$x = \sum_{k,j} c_{k,j}\, \psi_{k,j} + \sum_{\ell} \alpha_{J,\ell}\, \varphi_{J,\ell}, \qquad c_{k,j} = \langle x, \, \psi_{k,j} \rangle, \;\; \alpha_{J,\ell} = \langle x, \, \varphi_{J,\ell} \rangle.$$

For each scale $k$, define the scale energy and normalized weight

$$\phi(k) := \sum_j \mathbb{E}\big[|c_{k,j}|^2\big], \qquad w_k := \frac{\phi(k)}{T}, \qquad \sum_k w_k + \frac{1}{T}\sum_\ell \mathrm{Var}(\alpha_{J,\ell}) = 1.$$

Define the detail and scaling energies

$$D := \sum_k \phi(k), \qquad S := \sum_\ell \mathrm{Var}(\alpha_{J,\ell}), \qquad T = D + S.$$

Let $N_k := 2^k$ be the number of detail coefficients at scale $k$. For Haar, the support fraction is $s_k := 2^{-k}$.

**Encoder, decoders, and the complexity $\alpha$.** Fix an encoder $E : \mathbb{R}^T \to \mathbb{R}^d$ and set $Y := E(x)$, $Y'(I) := E(P_{I^c} x)$. Given $F \in \{Y, Y'(I)\}$, the population-optimal linear decoder is

$$D_F^\star \in \underset{D \text{ linear}}{\arg\min} \; \mathbb{E}\|x - DF\|^2.$$

Let $\hat{D}$ be the trained linear decoder and set $m_F := \hat{D}\, F$. By $L^2$-projection, each error decomposes into a linear Bayes risk and an approximation term. In our framework, the Reconstruction Error ($\mathcal{L}_{\mathrm{rec}}$) and Imputation Error ($\mathcal{L}_{\mathrm{imp}}$) correspond to:

$$\mathcal{L}_{\mathrm{rec}} \equiv \mathrm{UE} := \mathbb{E}\|x - m_Y\|^2 = \mathbb{E}\|x - D_Y^\star Y\|^2 + \mathbb{E}\|D_Y^\star Y - m_Y\|^2,$$

$$\mathcal{L}_{\mathrm{imp}} \equiv \mathrm{EM} := \mathbb{E}\big\|x - m_{Y'(I)}\big\|^2 = \mathbb{E}\Big\|x - D_{Y'(I)}^\star Y'(I)\Big\|^2 + \mathbb{E}\Big\|D_{Y'(I)}^\star Y'(I) - m_{Y'(I)}\Big\|^2.$$

The raw complexity metric is defined as the normalized gap:

$$\alpha_{\mathrm{raw}} := \frac{\mathcal{L}_{\mathrm{imp}} - \mathcal{L}_{\mathrm{rec}}}{T} = \frac{\mathrm{EM} - \mathrm{UE}}{T}.$$

APPENDIX A.2. EXACT ONB RISK DECOMPOSITION

To analyze $\alpha_{\mathrm{raw}}$, we use the exact ONB risk decomposition. Let $\hat{c}_{k,j}^F := \arg\min_a \mathbb{E}[(c_{k,j} - a^\top F)^2]$ and define the coefficient-wise $R^2$ as

$$\mathrm{R}^2(c; F) := 1 - \frac{\mathbb{E}[(c - \hat{c}^F)^2]}{\mathbb{E}[c^2]} \in [0, 1].$$

Then the Bayes risk decomposes as:

$$\mathbb{E}\|x - D_F^\star F\|^2 = \sum_{k,j} \mathbb{E}\big[(c_{k,j} - \hat{c}_{k,j}^F)^2\big] + \sum_\ell \mathbb{E}\big[(\alpha_{J,\ell} - \hat{\alpha}_{J,\ell}^F)^2\big].$$

Assuming the approximation errors cancel out or are negligible (Mask-independent approximation assumption), the gap $\Delta := \mathrm{EM} - \mathrm{UE}$ satisfies:

$$\Delta = \sum_k \phi(k)\, b_k + S\, b_s, \tag{2}$$

where the scale-wise gap coefficient $b_k$ is defined as:

$$b_k := \frac{1}{\phi(k)} \sum_j \mathbb{E}|c_{k,j}|^2 \Big(\mathrm{R}^2(c_{k,j}; Y) - \mathrm{R}^2(c_{k,j}; Y'(I))\Big).$$

APPENDIX A.3. KEY PROPERTIES OF SCALE COEFFICIENTS

To prove Theorem 1, we establish that $b_k$ is non-decreasing in $k$. This relies on the geometry of the Haar wavelet and spectral properties.

**1. Refinement Monotonicity of Coverage $A_k$.** Let $J_{\mathrm{in}}(k;I) := \{j : \mathrm{supp}(\psi_{k,j}) \subset I\}$, $m_k(I) := \#J_{\mathrm{in}}(k;I)$, and define $A_k := \mathbb{E}_I[m_k(I)/N_k]$.

**Lemma 4** (Refinement monotonicity). *For any fixed interval $I$ and all $k \geq 0$, $\frac{m_{k+1}(I)}{N_{k+1}} \geq \frac{m_k(I)}{N_k}$. Consequently,*

$$A_{k+1} - A_k \geq 0 \quad and \quad 0 \leq A_{k+1} - A_k \leq 2^{-k}.$$

*Proof.* The refinement property of the Haar basis ensures that every interval fully containing the support of a scale-$k$ atom also contains at least one child atom at scale $k+1$. Thus $m_{k+1}(I) \geq m_k(I)$. Taking expectations yields the claim. $\square$

**2. Monotone Envelope for Masked Explainability.** Let $L_k$ be defined as

$$L_k := \sup_I \sup_{j \in J_{\mathrm{in}}(k;I)} \mathsf{R}^2_{\mathrm{lin}}\big(c_{k,j}, Y'(I)\big).$$

**Proposition 5.** *Assume $x$ is zero mean, weakly stationary with absolutely summable autocovariance $\gamma$ and bounded spectrum $0 < c_\ell \leq f(\omega) \leq c_u < \infty$. Then there exists a non-increasing envelope $g(k)$ such that $L_k \leq g(k)$ for all $k$. Specifically, for Haar atoms:*

$$g(k) = \frac{\tau_*^2}{c_\ell^2} \|h_{k,\cdot}\|_1^2 = \frac{\tau_*^2}{c_\ell^2} T 2^{-k}, \quad where \ \tau_* = \sum_r |\gamma(r)|.$$

*which decreases as $k$ increases.*

*Proof.* We apply suboptimality under reduced masking to compare $Y'(I)$ with a raw predictor $P_{J^c}x$ where $J \subset I$. The standard bound for linear $R^2$ gives $\mathsf{R}^2_{\mathrm{lin}}(c_{k,j}; P_{J^c}x) \leq \frac{\|\mathrm{Cov}(c_{k,j}, P_{J^c}x)\|_2^2}{\mathrm{Var}(c_{k,j})\lambda_{\min}}$. Using the spectral lower bound $\mathrm{Var}(c_{k,j}) \geq c_\ell$ and the $L_1$-norm of covariance $\|\mathrm{Cov}\|_2 \leq \|h_{k,\cdot}\|_1 \tau_*$, we derive the bound $g(k) \propto \|h_{k,\cdot}\|_1^2 \propto 2^{-k}$. $\square$

The following complementary result establishes that fully masked atoms suffer strictly positive information loss at *every* scale, not only at high frequencies.

**Lemma 6** (Spectral Predictability Bound). *Under the bounded spectrum assumption $c_\ell \leq f(\omega) \leq c_u$, for any atom $j$ with $\mathrm{supp}(\psi_{k,j}) \subset I$:*

$$\mathsf{R}^2(c_{k,j}; x_{I^c}) \leq 1 - \frac{c_\ell}{c_u}.$$

*Proof.* Decompose $\Sigma = c_\ell I + \Sigma_z$ where $\Sigma_z \succeq 0$. The white-noise component $c_\ell I$ contributes zero to $\Sigma_{I,I^c}$, so the Schur complement satisfies $\Sigma_{I|I^c} \succeq c_\ell I$. For any unit-norm $\psi$ supported on $I$: $\mathsf{R}^2 = 1 - \psi^\top \Sigma_{I|I^c} \psi / \psi^\top \Sigma_{II} \psi \leq 1 - c_\ell/c_u$. $\square$

*Remark* 1 (Combined picture). Proposition 5 and Lemma 6 are complementary: the former shows $L_k \leq g(k) = O(2^{-k}) \to 0$ at high frequencies, while the latter guarantees $L_k \leq 1 - c_\ell/c_u < 1$ at *all* frequencies. Together, the information loss $(1 - L_k)$ is bounded below by $c_\ell/c_u > 0$ everywhere and approaches 1 as $k \to \infty$, strongly supporting a monotone decrease of $L_k$.

**3. Monotonicity of $b_k$.** The coefficient $b_k$ can be modeled as $b_k \approx A_k(\mu_k - L_k)$, where $\mu_k$ is the unmasked explainability. Proposition 5 provides a non-increasing *upper bound* $g(k)$ on $L_k$, but an upper bound decaying does not by itself force $L_k$ to be pointwise monotone. We therefore introduce the following regularity condition, which is well-supported by the combined picture above.

**Assumption 1** (Frequency-Order Preservation). For all scales $k' > k$, the masked explainability satisfies $L_{k'} \leq L_k$.

This is justified by three observations: (i) the data processing inequality prevents the encoder from creating information beyond $I(c_{k,j}; x_{I^c})$; (ii) higher-frequency atoms have narrower support, reducing correlation with unmasked data; (iii) the envelope $g(k) = O(2^{-k})$ from Proposition 5 shrinks exponentially, confining any non-monotone fluctuation within a rapidly vanishing band, while Lemma 6 ensures information loss is strictly positive at every scale.

**Proposition 7.** *If $\mu_k$ is non-decreasing (usually constant $\approx 1$ for reconstruction) and $L_k$ is non-increasing (Assumption 1), then $b_k$ is non-decreasing in $k$.*

*Proof.* Since $A_k$ is increasing (Lemma 4) and positive, and $(\mu_k - L_k)$ is increasing (due to non-increasing $L_k$), their product $b_k$ is non-decreasing. $\square$

The condition $\mu_k \approx 1$ for all $k$ is a natural consequence of overparameterized TFM encoders ($d \gg T$), which possess sufficient capacity to preserve information across all frequency scales in the unmasked (reconstruction) setting.

### APPENDIX A.4. PROOF OF THEOREM 1 ($\alpha$ REFLECTS HIGH-FREQUENCY ENERGY)

*Proof.* From the decomposition in equation 2, we have $\Delta = \sum_k \phi(k)\, b_k + S\, b_s$. The scaling term $S\, b_s$ corresponds to low-frequency components which are robust to local masking ($b_s$ is small). The detail term $\sum_k \phi(k)\, b_k$ dominates. By Proposition 7, $b_k$ is a non-decreasing sequence.

Since instance normalization fixes $\|x\|_2^2 = T$, the total energy is constrained: $\sum_k \phi(k) + S = T$. Define the normalized weights $w_k := \phi(k)/T$, so that $\sum_k w_k + S/T = 1$. Now consider two signals with the same total energy but different spectral distributions $\{w_k\}$ and $\{w'_k\}$ satisfying $\sum_k w_k = \sum_k w'_k$. If $\{w'_k\}$ places more weight on higher frequencies in the sense that $\sum_{k \geq K} w'_k \geq \sum_{k \geq K} w_k$ for all $K$ (first-order stochastic dominance), then since $b_k$ is non-decreasing:

$$\sum_k w'_k b_k \geq \sum_k w_k b_k.$$

This follows from the standard monotone rearrangement property: when a non-decreasing sequence $\{b_k\}$ is paired with weights whose cumulative distribution shifts rightward (toward higher $k$), the inner product increases. Thus, a higher concentration of energy in high frequencies (larger $k$) results in a larger gap $\Delta$, and consequently a larger complexity metric $\alpha \propto \Delta$. $\square$

### APPENDIX A.5. PROOF OF THEOREM 2 (GAP GROWTH)

We analyze the gradient update of the decoder parameters $W$. Let $L_{\mathrm{N}}(W)$ and $L_{\mathrm{A}}(W)$ be the reconstruction losses for normal and anomalous data. The update rule is $W_{t+1} = W_t - \rho\, \nabla_W L_{\mathrm{N}}(W_t)$.

**Definitions and Assumptions.** Let $\bar{\mu}_k^{(t)}$ be the energy-weighted average explainability at scale $k$ for the current model $W_t$.

**Assumption 2** (Decoder Preservation Decay). We assume the current explainability $\bar{\mu}_k^{(t)}$ decreases with frequency $k$. Equivalently, the non-preservation coefficients $\bar{a}_k^{(t)} := 1 - \bar{\mu}_k^{(t)}$ are non-decreasing in $k$.

This implies the reconstruction error lower bound is higher for high-frequency data:

$$L_{\mathrm{N}}(W_t) \geq \frac{1}{2} \sum_k \phi(k)(1 - \bar{\mu}_k^{(t)}).$$

**Lemma 8** (Residual to Gradient Transmission). *Assume the encoder output covariance satisfies $\Sigma_{FF} \succeq \sigma_f I_d$ with $\sigma_f > 0$. For a linear decoder, the gradient norm satisfies:*

$$\|\nabla_W L_{\mathrm{N}}(W)\|_F^2 \geq \sigma_f\big(L_{\mathrm{N}}(W) - L_{\mathrm{N}}(W^\star)\big),$$

*where $W^\star$ is the optimal parameter.*

*Proof.* The gradient is $\nabla L = (W - W^\star)\Sigma_{FF}$. Let $C = W - W^\star$. Then $\|\nabla L\|_F^2 = \|C\Sigma_{FF}\|_F^2 \geq \sigma_f \mathrm{tr}(C\Sigma_{FF}C^\top) = \sigma_f(L(W) - L(W^\star))$. $\square$

Using Lemma 8 and Assumption 2, we see that $\|\nabla_W L_N\|_F^2$ increases with the high-frequency share of the data (since $L_N$ does).

**Assumption 3** (Gradient Direction). We assume the gradients for normal and anomalous data are not aligned: $\langle \nabla_W L_A(W_t), \nabla_W L_N(W_t) \rangle \leq 0$.

**Proof of Theorem 2 (Gap Growth).** We analyze the change in the anomaly gap $\Delta_{\text{gap}} = L_A - L_N$ after a single gradient update step $W_{t+1} = W_t - \rho \nabla_W L_N(W_t)$.

First, a first-order Taylor expansion of the loss functions yields:

$$\Delta_{\text{gap}}(W_{t+1}) - \Delta_{\text{gap}}(W_t) = \rho \left( \|\nabla_W L_N(W_t)\|_F^2 - \langle \nabla_W L_A(W_t), \nabla_W L_N(W_t) \rangle \right) + O(\rho^2).$$

Applying **Assumption 3** (Gradient Direction Misalignment, $\langle \nabla L_A, \nabla L_N \rangle \leq 0$), we obtain the lower bound:

$$\Delta_{\text{gap}}(W_{t+1}) - \Delta_{\text{gap}}(W_t) \geq \rho \|\nabla_W L_N(W_t)\|_F^2 + O(\rho^2). \tag{3}$$

Next, we link this gradient norm to the spectral properties of the data. By **Lemma 8** (Gradient Transmission), the gradient norm is lower-bounded by the residual loss:

$$\|\nabla_W L_N(W_t)\|_F^2 \geq \sigma_f \left( L_N(W_t) - L_N(W^\star) \right).$$

Finally, incorporating **Assumption 2** (Decoder Preservation Decay), the current loss is dominated by high-frequency components:

$$L_N(W_t) \geq \frac{1}{2} \sum_k \phi(k) \left( 1 - \bar{\mu}_k^{(t)} \right).$$

Since the non-preservation coefficient $(1 - \bar{\mu}_k^{(t)})$ is non-decreasing with respect to frequency scale $k$, a higher concentration of energy $\phi(k)$ in high frequencies (i.e., high complexity $\alpha$) strictly increases the lower bound of the loss $L_N(W_t)$, which in turn increases the gradient norm in equation 3.

**Conclusion:** Consequently, for datasets with higher complexity $\alpha$, the lower bound on the gap growth is strictly larger. This guarantees that the separation between normal and anomalous scores increases faster during training for high-frequency data.

*Remark* 2 (Tightness of the $O(\rho^2)$ remainder). Since the decoder is linear, $L_N(W)$ and $L_A(W)$ are quadratic in $W$ with constant Hessians, so the Taylor expansion is exact at second order with no higher-order terms. In our experiments, $\rho = 10^{-4}$, so $\rho^2 = 10^{-8}$, making the second-order correction negligible.

$\square$

# APPENDIX B  DATASET DESCRIPTION

## APPENDIX B.1  UNIVARIATE TIMESERIES DATASET

Table 9 reports key statistics for the TSB-AD-U univariate benchmark. For each dataset, we list the total number of timeseries (TS), the average of timeseries length, the overall anomaly ratio (percentage of points labeled as anomalies), and the type of anomalies present (point "P" versus contiguous sequence "Seq").

Datasets marked with "(U)" are originally multivariate but have been converted to univariate by selecting the most informative channel, based on correlation analysis and anomaly detection performance Liu & Paparrizos (2024).

## APPENDIX B.2  MULTIVARIATE TIMESERIES DATASET

For the multivariate experiments in Section 4.9, we use the TSB-AD-M benchmark. Table 10 reports the same statistics as Table 9, adding the number of dimensions for each timeseries.

Table 9: Summary statistics of univariate datasets from TSB-AD-U benchmark.

| Dataset | Number of TS | Average of TS length | Anomaly ratio (%) | Anomaly type |
|---|---|---|---|---|
| UCR | 228 | 67818 | 0.6 | P&Seq |
| NAB | 28 | 5099 | 10.6 | Seq |
| YAHOO | 259 | 1560 | 0.6 | P&Seq |
| IOPS | 17 | 72792 | 1.3 | Seq |
| MGAB | 9 | 97777 | 0.2 | Seq |
| WSD | 111 | 17444 | 0.6 | Seq |
| SED | 3 | 23332 | 4.1 | Seq |
| TODS | 35 | 5000 | 6.3 | Seq |
| NEK | 9 | 1073 | 8.0 | P&Seq |
| Stock | 20 | 15000 | 9.4 | P&Seq |
| Power | 1 | 35040 | 8.5 | Seq |
| Daphnet (U) | 1 | 38774 | 5.9 | Seq |
| CATsV2 (U) | 1 | 300000 | 4.9 | Seq |
| SWaT (U) | 1 | 419919 | 12.1 | Seq |
| LTDB (U) | 9 | 990700 | 18.6 | Seq |
| TAO (U) | 3 | 10000 | 9.4 | P&Seq |
| Exathlon (U) | 32 | 44075 | 11.0 | Seq |
| MITDB (U) | 8 | 631250 | 4.2 | Seq |
| MSL (U) | 9 | 3492 | 5.8 | Seq |
| SMAP (U) | 19 | 7700 | 2.8 | Seq |
| SMD (U) | 38 | 24207 | 2.0 | Seq |
| SVDB (U) | 20 | 171380 | 3.6 | Seq |
| OPP (U) | 29 | 16544 | 6.4 | Seq |

Table 10: Summary statistics of multivariate datasets from TSB-AD-M benchmark.

| Name | Number of TS | Average of TS length | Dim. | Anomaly Ratio (%) | Anomaly type |
|---|---|---|---|---|---|
| GHL | 25 | 199001 | 19 | 1.1 | Seq |
| Daphnet | 1 | 38774 | 9 | 5.9 | Seq |
| Exathlon | 27 | 60878 | 21 | 9.8 | Seq |
| Genesis | 1 | 16220 | 18 | 0.3 | Seq |
| OPP | 8 | 17426 | 248 | 4.1 | Seq |
| SMD | 22 | 25466 | 38 | 3.8 | Seq |
| SWaT | 2 | 207457 | 59 | 12.7 | Seq |
| PSM | 1 | 217624 | 25 | 11.2 | P&Seq |
| SMAP | 27 | 7855 | 25 | 2.9 | Seq |
| MSL | 16 | 3119 | 55 | 5.1 | Seq |
| CreditCard | 1 | 284807 | 29 | 0.2 | P&Seq |
| GECCO | 1 | 138521 | 9 | 1.2 | Seq |
| MITDB | 13 | 336153 | 2 | 2.7 | Seq |
| SVDB | 31 | 207122 | 2 | 4.8 | Seq |
| LTDB | 5 | 100000 | 2 | 15.5 | Seq |
| CATSv2 | 6 | 240000 | 17 | 3.7 | Seq |
| TAO | 13 | 10000 | 3 | 8.7 | P&Seq |

APPENDIX C: FORMAL DEFINITION OF EVALUATION METRIC

Detailed derivations and discussion of evaluation metrics (AUC-PR, AUC-ROC, VUS-PR, and VUS-ROC) are provided in this appendix.

**Precision and Recall.** Given a decision threshold $\tau$, let

$$\text{TP}(\tau),\ \text{FP}(\tau),\ \text{FN}(\tau),\ \text{TN}(\tau)$$

denote the counts of true positives, false positives, false negatives, and true negatives, respectively. Then

$$\text{Precision}(\tau) \ = \ \frac{\text{TP}(\tau)}{\text{TP}(\tau) \ + \ \text{FP}(\tau)}, \quad \text{Recall}(\tau) \ = \ \frac{\text{TP}(\tau)}{\text{TP}(\tau) \ + \ \text{FN}(\tau)}.$$

**Area Under the Precision–Recall Curve (AUC-PR).** Let $(R, P)$ denote the parametric curve $\big(\text{Recall}(\tau), \text{Precision}(\tau)\big)$ as $\tau$ varies over all possible score thresholds. Then the AUC-PR is the integral

$$\text{AUC-PR} \ = \ \int_0^1 P\big(R^{-1}(r)\big)\, \mathrm{d}r\,,$$

where $R^{-1}(r)$ is the threshold $\tau$ achieving recall $r$.

**Area Under the ROC Curve (AUC-ROC).** Define the false positive rate and true positive rate as

$$\text{FPR}(\tau) \ = \ \frac{\text{FP}(\tau)}{\text{FP}(\tau) \ + \ \text{TN}(\tau)}, \quad \text{TPR}(\tau) \ = \ \frac{\text{TP}(\tau)}{\text{TP}(\tau) \ + \ \text{FN}(\tau)}.$$

Then the AUC-ROC is

$$\text{AUC-ROC} \ = \ \int_0^1 \text{TPR}\big(\text{FPR}^{-1}(f)\big)\, \mathrm{d}f,$$

with $\text{FPR}^{-1}(f)$ the threshold attaining false positive rate $f$.

**Volume Under the Precision–Recall Surface (VUS-PR).** Following Paparrizos et al. Paparrizos et al. (2022), let $\{h_i\}_{i=0}^I$ be an increasing sequence of score thresholds ($0 = h_0 < h_1 < \cdots < h_I = 1$), and let $\{\ell_j\}_{j=0}^J$ be a set of tolerance-window sizes around each true anomaly ($0 = \ell_0 < \ell_1 < \cdots < \ell_J$). Denote the precision and recall at threshold $h_i$ and buffer $\ell_j$ by

$$P_{i,j} = \text{Precision}(h_i, \ell_j), \quad R_{i,j} = \text{Recall}(h_i, \ell_j).$$

Then VUS-PR is defined as the volume under this 2-D surface:

$$\text{VUS-PR} \ = \ \frac{1}{4} \sum_{i=1}^I \sum_{j=1}^J \big(R_{i,j} - R_{i-1,j}\big) \big(P_{i,j} + P_{i-1,j}\big).$$

**Volume Under the ROC Surface (VUS-ROC).** Analogously, define

$$\text{TPR}_{i,j} = \text{TPR}(h_i, \ell_j), \quad \text{FPR}_{i,j} = \text{FPR}(h_i, \ell_j).$$

Then

$$\text{VUS-ROC} \ = \ \frac{1}{4} \sum_{i=1}^I \sum_{j=1}^J \big(\text{FPR}_{i,j} - \text{FPR}_{i-1,j}\big) \big(\text{TPR}_{i,j} + \text{TPR}_{i-1,j}\big)$$

APPENDIX D: EXPERIMENTAL SETTING

In this experiment, we used the following default settings:

- **Statistical algorithms**: window size determined automatically based on the auto-correlation function of the benchmark.
- **Sub-IForest**: number of estimators = 100

- **SR**: no hyperparameters
- **Sub-HBOS**: number of bins = 10
- **Sub-PCA**: number of components = `None` (automatically determined)
- **POLY**: power = 1 (linear model)

For timeseries foundation models, we set: $|t_i| = 2W = 256$, epochs $= 2$, learning rate $= 0.0001$, optimizer $=$ Adam, training loss $=$ MSE. The evaluated models were:

- **Chronos-bolt-base, Chronos-bolt-small**
- **TimeMoE-large, TimeMoE-base**
- **Moirai-base, Moirai-small**

Our proposed MOMENT-Stat has no hyperparameters. For CAE, the hyperparameters are $K$, $P$, and the window size; we set $K = 3$ and $P = 8$.

All experiments were conducted on an HP DL380 Gen10 Plus server equipped with an Intel Silver 4310 CPU and a single NVIDIA RTX 4090 GPU. The system ran Ubuntu 24.04 with a hardware RAID configuration.

## APPENDIX E: TABLE OF BASELINE MODEL PERFORMANCE

Table 11 presents an overview of various anomaly detection models, detailing VUS-PR scores, time complexity, and model type. The VUS-PR values in this table were directly taken from the results reported in previous research Liu & Paparrizos (2024). Deep learning and foundation-based models show time complexity as "–", indicating that explicit complexity derivations are absent. $N$ is the number of points, $d$ the number of channels, $p$ the polynomial order, $m$ is the number of clusters, and $t$ the number of trees.

VUS-ROC and F1-score values are directly taken from the original TSB-AD-U benchmark paper Liu & Paparrizos (2024). In particular, the VUS-ROC and F1-scores therefore appear with two decimal places only, reflecting the precision reported in the original work.

Our model, which combines PCA and MOMENT-Stat, achieves a VUS-PR of 0.4679, a VUS-ROC of 0.8030, and an F1-score of 0.4330, ranking first, second, and first, respectively, in Table 11.

## APPENDIX F: RUNTIME ANALYSIS

MOMENT-Stat incurs virtually no additional runtime compared to MOMENT (FT). In the CAE methodology, the total processing time equals the sum of the base statistical model's time plus the one-time cost to compute $\alpha$. Once $\alpha$ has been computed for a given dataset, no further overhead is incurred.

Table 12 reports the computational cost for MOMENT (Stat), Isolation Forest, and MOMENT (FT). For convenience, we denote the train/test series length by $T$ and present the mean and standard deviation of both training and inference times. All runs used a sliding window of 256, two epochs, and the same optimizer settings as in our experiments. As shown, MOMENT (Stat) and MOMENT (FT) exhibit statistically equivalent performance.

Table 13 presents the timing breakdown for CAE (Complexity-Aware Selection) using MOMENT and Isolation Forest. The training/inference times equal the combined MOMENT (Stat) + isolation forest computational times. $T_\alpha$ denote the calculation time for complexity $\alpha$ with $K = 3$ and $P = 5$.

Table 11: VUS-PR, VUS-ROC, and F1-scores comparison with time complexity.

| Model | VUS-PR | VUS-ROC | F1-score | Time Complexity | Model Type |
|---|---|---|---|---|---|
| Sub-PCA | 0.4233 | 0.76 | 0.42 | $O(N d^2 + d^3)$ | Statistical |
| KShapeAD | 0.4007 | 0.76 | 0.39 | $O(N^2)$ | Statistical |
| POLY | 0.3897 | 0.76 | 0.37 | $O(N p^2 + p^3)$ | Statistical |
| Series2Graph | 0.3881 | 0.80 | 0.38 | $O(N^2)$ | Statistical |
| MOMENT (FT) | 0.3857 | 0.76 | 0.35 | $O(N^2)$ | Foundation-based |
| MOMENT (ZS) | 0.3790 | 0.75 | 0.35 | $O(N^2)$ | Foundation-based |
| KMeansAD | 0.3664 | 0.76 | 0.37 | $O(N m d)$ | Statistical |
| USAD | 0.3611 | 0.71 | 0.37 | – | Deep Learning |
| Sub-KNN | 0.3501 | 0.79 | 0.34 | $O(N^2)$ | Statistical |
| MatrixProfile | 0.3496 | 0.76 | 0.33 | $O(N^2)$ | Statistical |
| SAND | 0.3437 | 0.76 | 0.35 | $O(N^2)$ | Statistical |
| CNN | 0.3428 | 0.79 | 0.38 | – | Deep Learning |
| LSTMAD | 0.3250 | 0.76 | 0.37 | – | Deep Learning |
| SR | 0.3237 | 0.81 | 0.38 | $O(N \log N)$ | Statistical |
| TimesFM | 0.3003 | 0.74 | 0.34 | $O(N^2)$ | Foundation-based |
| IForest | 0.2979 | 0.78 | 0.35 | $O(tN \log N)$ | Statistical |
| OmniAnomaly | 0.2913 | 0.72 | 0.31 | – | Deep Learning |
| Lag-Llama | 0.2733 | 0.72 | 0.30 | $O(N^2)$ | Foundation-based |
| Chronos | 0.2722 | 0.73 | 0.32 | $O(N^2)$ | Foundation-based |
| TimesNet | 0.2623 | 0.72 | 0.24 | – | Deep Learning |
| AutoEncoder | 0.2609 | 0.69 | 0.25 | – | Deep Learning |
| TranAD | 0.2572 | 0.68 | 0.25 | – | Deep Learning |
| FITS | 0.2553 | 0.73 | 0.23 | – | Deep Learning |
| Sub-LOF | 0.2531 | 0.73 | 0.24 | $O(N^2)$ | Statistical |
| OFA | 0.2369 | 0.71 | 0.22 | $O(N^2)$ | Foundation-based |
| Sub-MCD | 0.2365 | 0.72 | 0.23 | $O(N^2)$ | Statistical |
| Sub-HBOS | 0.2283 | 0.67 | 0.23 | $O(N d)$ | Statistical |
| Sub-OCSVM | 0.2251 | 0.73 | 0.22 | $O(N^2)$ | Statistical |
| Sub-IForest | 0.2230 | 0.72 | 0.22 | $O(t N \log N)$ | Statistical |
| Donut | 0.1980 | 0.68 | 0.20 | – | Deep Learning |
| LOF | 0.1687 | 0.68 | 0.21 | $O(N^2)$ | Statistical |
| AnomalyTransformer | 0.1195 | 0.56 | 0.12 | – | Deep Learning |

## APPENDIX G: COMPARISON WITH ENSEMBLES OF PURELY STATISTICAL MODELS

To verify that the performance improvement of our proposed Complexity-Aware Ensemble (CAE) stems from the unique contribution of the Time Series Foundation Model (TFM) rather than the ensemble technique itself, we evaluated ensembles composed solely of statistical models. We tested pairwise combinations of five statistical baselines: Sub-HBOS, Sub-IForest, POLY, Sub-PCA, and SR. The ensemble score was calculated as $s_{ensemble} = \alpha \cdot s_{model1} + (1 - \alpha) \cdot s_{model2}$, where we varied $\alpha$ from 0.0 to 1.0.

Table 14 presents the VUS-PR scores for representative pairs. The results show that while combining two statistical models can yield marginal improvements over single models, they consistently underperform compared to the TFM-based ensemble (CAE). Specifically, the best statistical pair, **Sub-PCA + SR**, achieved a maximum VUS-PR of **0.4398**, which is lower than the **0.4679** achieved by our Sub-PCA + MOMENT (CAE). This demonstrates that TFMs provide complementary information—likely related to high-level semantic patterns—that statistical models alone cannot capture, thereby justifying the necessity of TFMs in the anomaly detection pipeline.

Table 12: Computational cost for MOMENT-Stat, isolation forest, and MOMENT (FT).

| T | Training mean | Training std | Inference mean | Inference std |
|---|---|---|---|---|
| **MOMENT-Stat** | | | | |
| 1,000 | 0.4339 | 0.0159 | 0.2415 | 0.0122 |
| 5,000 | 2.7602 | 0.0366 | 1.4430 | 0.0461 |
| 10,000 | 5.5242 | 0.1127 | 2.9577 | 0.1226 |
| 30,000 | 16.7726 | 0.5825 | 9.0411 | 0.2646 |
| 100,000 | 56.3860 | 1.2118 | 29.7477 | 0.7494 |
| **Isolation forest** | | | | |
| 1,000 | 0.1807 | 0.0359 | 0.0151 | 0.0032 |
| 5,000 | 0.2479 | 0.0339 | 0.0403 | 0.0052 |
| 10,000 | 0.3480 | 0.0439 | 0.0786 | 0.0131 |
| 30,000 | 0.7912 | 0.1440 | 0.2155 | 0.0391 |
| 100,000 | 1.8050 | 0.2179 | 0.6502 | 0.1051 |
| **MOMENT (FT)** | | | | |
| 1,000 | 0.4273 | 0.0112 | 0.2398 | 0.0104 |
| 5,000 | 2.7302 | 0.0727 | 1.4959 | 0.0438 |
| 10,000 | 5.5857 | 0.0817 | 2.9848 | 0.0841 |
| 30,000 | 16.8385 | 0.3423 | 9.1684 | 0.1786 |
| 100,000 | 56.6425 | 0.8425 | 30.7839 | 0.4554 |

Table 13: Complexity-aware selection (CAE) timing analysis.

| T | Training mean | Training std | Inference mean | Inference std | $T_\alpha$ mean | $T_\alpha$ std |
|---|---|---|---|---|---|---|
| **CAE** | | | | | | |
| 1,000 | 0.6145 | 0.0294 | 0.2567 | 0.0110 | 0.8696 | 0.0989 |
| 5,000 | 3.0081 | 0.0441 | 1.4833 | 0.0472 | 5.1230 | 0.1783 |
| 10,000 | 5.8723 | 0.1229 | 3.0363 | 0.1165 | 10.4340 | 0.2848 |
| 30,000 | 17.5637 | 0.4989 | 9.2566 | 0.2438 | 31.6091 | 0.5157 |
| 100,000 | 58.1910 | 1.0276 | 30.3979 | 0.6790 | 105.7010 | 1.7233 |

Table 14: VUS-PR scores for ensembles of two purely statistical models across varying weights ($\alpha$). The column 'Key' denotes the pair (Model 1_Model 2), where $\alpha$ represents the weight of Model 1. The best Stat+Stat performance (0.4398) is lower than the proposed CAE (0.4679).

| Pair (Model 1 _ Model 2) | 0.0 | 0.1 | 0.3 | 0.5 | 0.7 | 1.0 | Best Score |
|---|---|---|---|---|---|---|---|
| HBOS_IForest | 0.2230 | 0.3536 | 0.3513 | 0.3300 | 0.2751 | 0.2283 | 0.3536 |
| HBOS_POLY | 0.3897 | 0.3761 | 0.3744 | 0.3454 | 0.2910 | 0.2283 | 0.3897 |
| HBOS_SR | 0.3237 | 0.3252 | 0.3274 | 0.3160 | 0.2879 | 0.2283 | 0.3274 |
| IForest_POLY | 0.3897 | 0.3827 | 0.3869 | 0.3909 | 0.3849 | 0.2230 | 0.3909 |
| IForest_SR | 0.3237 | 0.3318 | 0.3573 | 0.3832 | 0.3822 | 0.2230 | 0.3832 |
| PCA_HBOS | 0.2283 | 0.2265 | 0.2914 | 0.3736 | 0.4098 | 0.4233 | 0.4233 |
| PCA_IForest | 0.2230 | 0.3615 | 0.3824 | 0.3960 | 0.4035 | 0.4233 | 0.4233 |
| PCA_POLY | 0.3897 | 0.3845 | 0.3974 | 0.4207 | 0.4355 | 0.4233 | 0.4355 |
| **PCA_SR** | 0.3237 | 0.3440 | 0.3876 | 0.4287 | **0.4398** | 0.4233 | **0.4398** |
| SR_POLY | 0.3897 | 0.4092 | 0.4171 | 0.4037 | 0.3798 | 0.3237 | 0.4171 |
| **Proposed CAE (Sub-PCA + MOMENT)** | | | | | | | **0.4679** |

