# OpenReview forum: "Complexity- and Statistics-Guided Anomaly Detection in Time Series Foundation Models"
_ICLR.cc/2026/Conference — ICLR 2026 Poster_

### Official Review · Reviewer_q2N4 · 2025-10-31

**Soundness:** 2
**Presentation:** 2
**Contribution:** 3
**Rating:** 4
**Confidence:** 4

**Summary:**

This paper aims to address the issues of overgeneralization and overstationarity in Time Series Foundation Models (TSFMs). To mitigate overgeneralization, it introduces an adaptive weighting mechanism that combines anomaly scores from TSFMs and traditional statistical methods. To alleviate overstationarity, the model integrates mean and variance information into the encoder’s input representation.

**Strengths:**

1. The paper tackles the overgeneralization issue, which is a very important yet insufficiently addressed problem in anomaly detection using Time Series Foundation Models (TSFMs).
2. The introduction is well-written, clearly presenting the motivation, the research problem, and the methodological approach, making it easy for readers to follow the authors’ reasoning.
3. The experimental results convincingly demonstrate that TSFMs indeed suffer from the overgeneralization problem, providing strong empirical support for the paper’s premise.

**Weaknesses:**

1. The paper’s notation system is somewhat confusing, and the methodological presentation could be improved. For example, in lines 153–156, several symbols with different subscripts (e.g., P with varying indices) appear without clear definitions, making the section difficult to follow. Moreover, key background concepts such as scale energy and detail energy should be introduced in a dedicated Preliminaries section to help readers better understand the core methodological design.

2. While the paper points out that TSFMs suffer from overstationarization, it lacks further discussion or empirical evidence on how normalization operations affect anomaly detection accuracy.

3. The experimental setup is not fully convincing:
3.1. Since TSFMs exhibit more severe overgeneralization than time-series models specifically designed for anomaly detection, it is essential to compare the proposed CAE method not only with widely used TSFMs (e.g., One-Fits-All, Moment) but also with task-specific anomaly detection models. The current experiments do not sufficiently demonstrate that CAE outperforms existing anomaly detection methods.
3.2. To substantiate the claim that CAE alleviates overgeneralization, the paper should also compare the recall values of different methods (TSFMs and dedicated anomaly detection models such as Anomaly Transformer and DCDetector) under the condition of achieving their respective best F1 scores.

4. It is doubtful whether mainly combining the mean and variance with the input representation in the encoder and optimizing it by reconstruction error can solve the problem of overstationarity. The experiments on this question are also unconvincing as Moment-stat only makes a marginal improvement compared with Moment (FT).

**Questions:**

1. Could the authors provide both theoretical and experimental analyses to clarify how normalization operations affect anomaly detection performance?

2. I suggest comparing the proposed CAE with TSFMs such as One-Fits-All and Moment, as well as anomaly detection models like Anomaly Transformer and DCDetector, in terms of VUS-PR to more comprehensively evaluate its effectiveness.

3. I recommend adding a comparison of recall values under the condition of achieving the best F1 score for each method, to better demonstrate that the proposed approach effectively mitigates the overgeneralization problem.

4. Could the authors elaborate on the underlying principle of how incorporating mean and variance into the encoder input can effectively address the overstationarization problem?

---

> ### Author Response · Authors · 2025-11-21
>
> ## General response
>
> We thank the reviewer for the thoughtful review and constructive feedback. We appreciate the recognition of (i) the importance of overgeneralization and over-stationarization in TFMs, (ii) the clarity of our motivation, and (iii) the strength of our empirical evidence. **In the revision, we clarified the theorems behind our complexity metric $\alpha$, improved the notation in Section 3, and added multivariate and F1-score experiments.**
>
> &nbsp;
>
> ## W1. Notation and methodological clarity
>
> We agree that the original notation in Section 3 was hard to follow. In the revised manuscript, we
> (i) standardize the complexity metric to $\alpha$ and explicitly define $L_{rec}$ and $L_{imp}$,
> (ii) keep the main text focused on intuition and the overall CAE pipeline, and
> (iii) move detailed proofs and energy-related derivations to the appendix.
>
> &nbsp;
>
> ## W2. Effect of normalization on anomaly detection accuracy
>
> We added Lemma 3.3 in the revised version to formally analyze this effect. We show that instance normalization (e.g., RevIN), while helpful for forecasting in TFMs, has not previously been reported to harm anomaly detection, yet it can do so by discarding statistics that define some anomalies. In Section 3.3, Lemma 3.3 proves that a reconstruction model $f(x) = D(E(\mathcal{N}(x)))$ is invariant to affine transforms $x' = \alpha x + \beta$, so anomalies that only change mean or variance are invisible. To address this, MOMENT-Stat re-injects instance statistics into the decoder and improves VUS-PR from 0.3857 (MOMENT-FT) to 0.3913, raising the rank from 6 to 3 among 25 baselines.
>
> The added lemma is as follows:
>
> **Lemma 3.3 (Affine Invariance under Instance Normalization).**
> Let $\mathcal{N}(x) = (x - \mu_x)/\sigma_x$ be the instance normalization, and let $f(x) = D(E(\mathcal{N}(x)))$ be the reconstruction model composed of an encoder \(E\) and decoder \(D\). For any input time series \(x\) and its affine transformation $x' = \alpha x + \beta$ (with $\alpha > 0$), the normalized inputs are identical:
> $$
> \mathcal{N}(x') = \mathcal{N}(x).
> $$
> Consequently,
> $$
> f(x') = D(E(\mathcal{N}(x'))) = D(E(\mathcal{N}(x))) = f(x).
> $$
>
> &nbsp;
>
> ## W3. Baselines: TSFMs vs task-specific anomaly detectors
>
> Our evaluation is built on the TSB-AD benchmark, which contains **25 baselines**, including TFMs (MOMENT FT/ZS, OFA, TimesFM, Chronos, Lag-Llama), deep anomaly detectors (Anomaly Transformer, TranAD, USAD), and statistical methods (Sub-PCA, SR, Matrix Profile, etc.). Under the benchmark’s tuned configurations, CAE (Sub-PCA + MOMENT) achieves VUS-PR 0.4679, outperforming all TFMs and task-specific detectors, including Sub-PCA (0.4233), Anomaly Transformer (0.1195), and OFA (0.2369).
>
> &nbsp;
>
> ## W4. Mean/variance injection and over-stationarization
>
> In the baseline model, the TFM with instance normalization does not receive the sample mean or sample variance at all, so it is inherently blind to anomalies that manifest only through level or scale changes. In our MOMENT-Stat, we explicitly provide these statistics to the decoder, so the TFM can at least use this information when reconstructing the signal. To the best of our knowledge, our work is the first to point out that over-stationarization in TFMs can harm anomaly detection, and we view our modification as a simple but meaningful step toward addressing this issue.
>
> &nbsp;
>
> ## Q1. Normalization operations and anomaly detection performance
> As summarized above (W2), Lemma 3.3 shows that instance normalization makes the encoder representation invariant to mean and variance, which explains the over-stationarization issue.
>
> &nbsp;
>
> ## Q2. Comparison with additional TSFMs and anomaly detectors in VUS-PR
> Appendix E (Table 11) already includes TFMs such as OFA (VUS-PR 0.2369) and deep anomaly detectors such as Anomaly Transformer (0.1195), along with many other baselines. Our CAE reaches VUS-PR 0.4679 and sets a new state of the art on TSB-AD, supporting the claim that combining TFMs with statistical models via CAE is beneficial.
>
> &nbsp;
>
> ## Q3. Recall at best F1
> Beyond AUC-type metrics, we compare methods at their best F1-score. CAE achieves the highest Best F1-score (0.4330), while Sub-PCA achieves 0.42 and Anomaly Transformer 0.12. This confirms that CAE improves both ranking quality (VUS-PR/VUS-ROC) and thresholded detection performance, which is closely related to overgeneralization.
>
> &nbsp;
>
> ## Q4. Principle behind mean/variance injection for over-stationarization
> In the baseline model, a TFM with instance normalization never sees the sample mean or variance, so it is inherently blind to anomalies that appear only as level or scale changes. In MOMENT-Stat, we keep the encoder shape-focused by feeding it only the normalized input, but explicitly pass $(\mu,\sigma)$ to the decoder. This way, the reconstruction can depend on both shape and original statistics, partially correcting over-stationarization in a simple and principled manner.

---

> > ### Comment · Reviewer_q2N4 · 2025-11-26
> >
> > Thank you for your response. I appreciate the effort of the authors during the rebuttal process. Thus, I raise my score accordingly.

---

> > > ### Author Response · Authors · 2025-11-27
> > >
> > > We appreciate your time and the re-evalation of our paper. We are delighted that our revisions addressed your feedbacks, and thank you for the rasing score.

---

### Official Review · Reviewer_aW8c · 2025-10-31

**Soundness:** 3
**Presentation:** 3
**Contribution:** 3
**Rating:** 6
**Confidence:** 3

**Summary:**

In this paper, the authors propose an anomaly detection method built on time-series foundation models (TFMs). The authors found that when using TFMs (pre-trained for forecasting) in a reconstruction-based anomaly detection setting, the model may overgeneralize. Also, the use of instance normalization layers removes statistical features that are actually useful for anomaly detection. To address these issues, the authors propose a complexity metric  that measures how difficult a segment is for the TFM. Also, a method to reintroduce lost statistical features into the reconstruction process without retraining the TFM is introduced. Comprehensive experiments are conducted on 23 univariate benchmark datasets

**Strengths:**

- The use of foundation models (TFMs) in time‐series is growing, but the paper identifies important pitfalls when using them for anomaly detection
- The introduction of the complexity metric α and the ensemble approach is novel
- The authors run experiments on 23 univariate time-series datasets

**Weaknesses:**

- The experiments are only on univariate time series. Many real-world anomaly detection tasks are multivariate. The method’s applicability to multivariate TFMs is unclear. The paper does not sufficiently discuss how the method scales to multivariate or high-dimensional data.
- While the idea of a “complexity” metric is intuitive, the exact definition and properties of α need to be clearly explained. What does “complexity” measure exactly?
- How sensitive are results to the choice of α.
- Stronger empirical evidence regarding the claim that  "instance normalization (and other normalization) removes statistical features useful for anomaly detection" is needed.
- Reintroducing statistics is consistent across models and domains?
- If the underlying TFM is trained for forecasting, reconstruction may not be the most natural anomaly detection mechanism. Predictio-based anomaly detection might be better in some cases.

**Questions:**

- How are anomalies labelled? Are there multiple anomaly types per dataset?
- Are anomalies artificially injected or real events?
- Does the complexity metric α help provide insight and is that interpretably presented?

---

> ### Author Response · Authors · 2025-11-21
>
> ## General response
>
> We thank the reviewer for the careful reading and constructive feedback. We also appreciate the positive remarks on (i) identifying important pitfalls of TFMs for anomaly detection (S1), (ii) the novelty of the complexity metric $\alpha$ and the ensemble framework (S2), and (iii) the broad experimental coverage on 23 univariate datasets (S3). In this revision, we conduct experiments on Tmultivariate datasets. Furthermore, we have clarified Theorems 3.1 and 3.2 to provide a clearer and more intuitive explanation of the complexity metric $\alpha$.
>
> &nbsp;
>
> ## W1. Only univariate experiments
> We agree that multivariate anomaly detection is crucial. Section 4.9 now evaluates CAE on six multivariate datasets from the TSB-AD-M benchmark.
>
> As shown in Table 1 below, our CAE consistently outperforms the naive ensemble ($\alpha=0.5$) in terms of VUS-PR  across all statistical models that supprots multivariate data.
> **Table 1. Comparison between naive ensemble and CAE on six multivariate datasets.**
>
> | **Detector** | **VUS-PR ($\alpha=0.5$)** | **VUS-PR CAE** | **VUS-ROC ($\alpha=0.5$)** | **VUS-ROC CAE** |
> |-------------|---------------------------|----------------|----------------------------|-----------------|
> | HBOS        | 0.1751                    | **0.2535**     | 0.7099                     | **0.7324**      |
> | IForest     | 0.2553                    | **0.2996**     | **0.8070**                 | 0.7981          |
> | LOF         | 0.1091                    | **0.2122**     | 0.6389                     | **0.6796**      |
> | PCA         | 0.3132                    | **0.3433**     | **0.7834**                 | 0.7730          |
>
> Since MOMENT treats channels independently, it shows suboptimal performance on multivariate data. Consequently, an ensemble can yield worse results than using the statistical model alone, as documented in Table 7 of the revised manuscript.
>
> &nbsp;
>
> ## W2. Definition and properties of $\alpha$
> Section 3.2 defines $\alpha$ explicitly. For each segment $x$ we compute
> $$
> w(x) = \frac{L_{\mathrm{imp}}(x) - L_{\mathrm{rec}}(x)}{\lVert x \rVert_2^2}
> $$
> and map it to $[0,1]$ with a quantile transform. We then use $\alpha(x)$ directly as the ensemble weight between the TFM and statistical scores.
>
> Theoretically, we ground this metric in two key theorems:
>
> **Theorem 3.1**: Establishes that $\alpha$ is proportional to the high-frequency spectral energy of the time series, making it a rigorous measure of signal complexity.
>
> **Theorem 3.2**: Proves that a higher $\alpha$ leads to a larger gradient norm during optimization, which mathematically guarantees a wider separation (gap) between normal and anomalous scores.
>
> &nbsp;
>
> ## W3. Sensitivity of $\alpha$
>
> For sensitivity, Table 3 scans fixed $\alpha \in \{0, 0.1, \dots, 1\}$ and shows that using instance-wise $\alpha(x)$ matches or outperforms the best constant $\alpha$ for all backbones, and Table 4 shows that replacing our $\alpha$ with standard complexity measures consistently reduces VUS-PR.
>
> &nbsp;
>
> ## W4. Effect of normalization / instance normalization
>
> Lemma 3.3 shows that a TFM with instance normalization cannot react to anomalies that change only the mean or variance. Table 2 shows that our modification improves VUS-PR from $0.3857$ to $0.3913$ (6th to 3rd among 25 baselines), which is meaningful since the TFM now explicitly receives mean and variance as inputs.
>
> &nbsp;
>
> ## W5. Forecasting vs reconstruction-based anomaly detection
>
> We agree that prediction-based anomaly detection is natural for TFMs trained specifically for forecasting. Our main study focuses on reconstruction with MOMENT because it is explicitly trained for masked reconstruction. To address forecasting-based TFMs, Section 4.8 evaluates Chronos, TimeMoE, and Moirai using forecasting error as the anomaly score; lower sMAPE is strongly correlated with higher VUS-PR.
>
> We did not apply CAE to this setting, as the overgeneralization phenomenon we target appears primarily in reconstruction tasks.
>
> &nbsp;
>
> ## Q1. Anomaly labels and types
>
> Our experiments are conducted on a diverse collection of datasets that encompass three distinct anomaly types and a mix of real-world and synthetic data. As detailed in Appendix B (Tables 9–10), the datasets are annotated as having point anomalies (P), sequence anomalies (Seq), or both (P&Seq). This collection also combines real-world data with synthetically injected anomalies.
>
> &nbsp;
>
> ## Q2. Interpretability of $\alpha$
>
> By construction, $\alpha(x)$ measures how hard masked prediction $L_{\mathrm{imp}}$ is relative to full reconstruction $L_{\mathrm{rec}}$. When $L_{\mathrm{imp}}$ is small, the series is easy (simple). Consistent with Theorem 3.1, a larger error in masked regions indicates a higher density of local high frequency energy.
>
> As shown in Section 4.3 (Figure 2), simple linear trends yield a very low score ($\alpha \approx 0.0660$) whereas complex  signals yield a high score ($\alpha \approx 0.4053$).

---

### Official Review · Reviewer_zNh6 · 2025-11-01

**Soundness:** 2
**Presentation:** 2
**Contribution:** 2
**Rating:** 2
**Confidence:** 4

**Summary:**

This paper proposes a framework that leverages complexity- and statistics-guided ensembling (CAE) to improve the anomaly detection capability of TFMs. The approach aims to mitigate the overgeneralization and overstationarization issues commonly observed in pretrained TFMs by integrating a statistical anomaly detector with foundation-model representations.

**Strengths:**

* The paper is among the first to systematically investigate the use of time-series foundation models for anomaly detection tasks.
* The proposed complexity- and statistics-guided ensemble framework is well-motivated.

**Weaknesses:**

* Some claims of effectiveness, particularly regarding the superiority of TFMs within the CAE framework, are insufficiently supported by experiments.
* The evaluation scope is limited to univariate cases and reconstruction-based TFMs.

Please find the detailed comments in the following section.

**Questions:**

* The ensemble strategy indeed improves performance, but the claim that this improvement demonstrates the effectiveness of TFMs is not fully convincing. It is important to support the argument by comparing against ensembles of two purely statistical methods, thereby isolating the specific contribution of TFMs.
* The paper mentions overgeneralization and overstationarization issues, which also affect forecasting-based TFMs. Why are the proposed methodologies not extended or tested in this setting?
* The discussion on the multivariate case is incomplete. Could the CAE framework be generalized to such case?
* The difference between CAE and CAE-per-data should be more clearly articulated. What causes CAE-per-data to underperform, and what insights does this provide about model generalization?
* How is the ensemble factor w determined? Have the authors conducted a sensitivity analysis to assess its stability across datasets?
* Including more visual examples or case studies would improve readability and help contextualize the motivation behind the method.

---

> ### Author Response · Authors · 2025-11-21
>
> ## General response
>
> We thank the reviewer for the careful review and constructive feedback, and for recognizing both our investigation of TFMs for anomaly detection (S1) and the motivation of the CAE framework (S2). In the revision, we ran additional experiments with a purely statistical ensemble baseline and in multivariate settings, and clarified Theorems 3.1 and 3.2 to make explicit that $\alpha$ measures data complexity and to formalize its relationship with the anomaly score.
>
> &nbsp;
>
> ## W1. Effectiveness of TFMs within CAE
>
> We agree that our current study focuses on univariate, reconstruction-based settings.
>
> - **Multivariate extension.** We now report multivariate results on six datasets from TSB-AD-M (Section 4.9). CAE still brings clear gains. I will elaborate on this point in my response to Q3.
>
> - **Focus on reconstruction based TFMs.** Existing TFM based anomaly detection methods such as UniTS and MOMENT are reconstruction based, so in this work we concentrate on this setting.
>
> &nbsp;
>
> ## Q1. Ensemble vs purely statistical methods
>
> Following the suggestion, we compared CAE to purely statistical ensembles. We evaluated all pairwise ensembles of the five statistical baselines. The best statistical ensemble, Sub-PCA + SR, reaches VUS-PR $0.4398$. Our CAE with Sub-PCA + MOMENT attains VUS-PR $0.4679$, while Sub-PCA alone has $0.4233$. These results (Appendix G) show that TFMs provide complementary information that purely statistical ensembles cannot match.
>
> &nbsp;
>
> ## Q2. Extension to forecasting based TFMs
>
> While a forecasting specific CAE is an interesting direction, the overgeneralization phenomenon we target, namely degraded performance on simple datasets, appears only in reconstruction settings. Our experiments therefore suggest that for forecasting TFMs, as in Section 4.8, improving predictive accuracy is currently a more direct way to enhance anomaly detection performance than applying CAE.
>
> &nbsp;
>
> ## Q3. Multivariate case
>
> For each statistical detector that supprots multivariate data, we consider three variants: the pure statistical model, a naive ensemble with fixed $\alpha=0.5$, and CAE. Because TFMs such as MOMENT use channel independent decoding, they can be weaker than multivariate methods like PCA on their own. Nonetheless, CAE improves or matches the best component in most cases. For instance, CAE improves VUS-PR from $0.1751$ to $0.2535$ for HBOS and from $0.1091$ to $0.2122$ for LOF, while remaining competitive for IForest and PCA. This indicates that the proposed ensemble method generalizes beyond the univariate setting.
>
> **Table 1. Comparison between naive ensemble and CAE on six multivariate datasets.**
>
> | **Detector** | **VUS-PR ($\alpha=0.5$)** | **VUS-PR CAE** | **VUS-ROC ($\alpha=0.5$)** | **VUS-ROC CAE** |
> |-------------|---------------------------|----------------|----------------------------|-----------------|
> | HBOS        | 0.1751                    | **0.2535**     | 0.7099                     | **0.7324**      |
> | IForest     | 0.2553                    | **0.2996**     | **0.8070**                 | 0.7981          |
> | LOF         | 0.1091                    | **0.2122**     | 0.6389                     | **0.6796**      |
> | PCA         | 0.3132                    | **0.3433**     | **0.7834**                 | 0.7730          |
>
> &nbsp;
>
> ## Q4. CAE vs CAE-per-data
>
> The main difference is statistical stability. CAE uses a dataset-level estimator
>
> $$
> \bar{\alpha} = \frac{1}{N}\sum_{i=1}^{N} \hat{\alpha}_i,
> $$
>
> where $\hat{\alpha}_i$ is the complexity of the $i$-th series. This approximates
>
> $$
> \bar{\alpha} \approx \mathbb{E}_{X \sim \mathcal{P}}[\alpha(X)],
> $$
>
> which reduces variance and empirically yields better generalization than the per-series variant (CAE-per-data).
>
> &nbsp;
>
> ## Q5. Ensemble factor $\alpha$ and sensitivity
>
> We apologize for the confusing notation in the original draft. In the revision we consistently denote the ensemble factor by $\alpha$ and clarify its construction. We first compute
> $$ w(x) = \frac{L_{\mathrm{imp}}(x) - L_{\mathrm{rec}}(x)}{\lVert x \rVert_2^2}, $$
> apply a quantile transform to map it to $[0,1]$,
> $$ \alpha(x) = \mathrm{QuantileTransform}\bigl(w(x)\bigr), $$
> and use $\alpha(x)$ as the ensemble weight
> $$ s_{\mathrm{CAE}}(x) = \alpha(x)s_{\mathrm{TFM}}(x) + \bigl(1-\alpha(x)\bigr)s_{\mathrm{stat}}(x). $$
>
> To assess sensitivity, we compare CAE to ensembles with fixed $\alpha \in \{0.0,0.1,\dots,1.0\}$ (Table 3) and to variants that use alternative complexity measures such as spectral entropy and approximate entropy (Table 4). Across these experiments, our adaptive $\alpha$ performs better than the alternative measures.
>
> &nbsp;
>
> ## Q6. More visual examples and case studies
>
> We appreciate this suggestion. In this revision, we prioritized adding multivariate experiments and clarifying the theoretical development of the complexity metric and CAE, which we hope strengthens confidence in our methodology.

---

### Official Review · Reviewer_BRgs · 2025-11-01

**Soundness:** 2
**Presentation:** 3
**Contribution:** 2
**Rating:** 4
**Confidence:** 4

**Summary:**

The paper introduces methods which enhance the performance of time series foundation models (TFMs) in two different ways. The first is a direct adjustment on the networks so that global timeseries statistical information (mean, variance) which is lost in normalization steps is re-introduced in the end. The second method, called compexity-aware ensemble (CAE), ensembles the TFM with a statistical methods in an adaptive way to improve performance.

After a short description of the task, the CAE method and the statistical augmenting method applied on the RevIN layer are introduced. The TFM used is MOMENT, which was along the high scoring NN-based methods in the TSB-AD-U benchmark. When statistially enhanced, it is called MOMENT-Stat.

In the experiments section, the effect of CAE is demonstrated on different types of synthetic series. Then MOMENT-stat is compared with the original versions of MOMENT and shown to achieve slightly better scores. Afterwards, MOMENT-stat is fixed as the TFM used and CAE is applied on different statistical methods and shown to consistently increase their performance. An ablation is also present comparing different fixed values of alpha with CAE which results to the best performance. Additionally, alternative methods to weight the ensemble based on entropy are compared with CAE, again yielding lower performance. Finally, on an additional section some forecasting TFMs are briefly studied. It is observed that their forecasting performance is correlated with their anomaly detection performance and finally they are compared with each other and with MOMENT on multi-variate time series.

**Strengths:**

- The paper is quite readable and the method and goals are clear.

- Descent datasets and evaluation methods where used, something often missing in works on timeseries anomaly detection

- The paper touches different topics on TFMs and their usage in anomaly detection.

**Weaknesses:**

- It is not very clear from the paper that the CAE method is much better than simply ensembling statistical models with MOMENT. On table 3, one can see that the scores when using the value alpha = 0.5 are relatively close to the scores of CAE. To be convinced that CAE is preferable to simple ensembling one would expect a thorough analysis on how the values of alpha vary across datasets, where they peak and how close to this peak CAE is, compared to constant values of alpha.

- The relation of alpha to the ensembling weight w is not adequately explained. It is only mentioned that it is an increasing function. It is not clear if it is the same for all datasets or also adaptive or how it is defined. This also raises some more questions about the difference CAE with simple ensembling, as no information is present on e.g. how ensembles with fixed weights w (e.g. w = 0.5) would perform. It could be that those are close to the columns of table 3, but it is not clear. Finally, this makes it also difficult to judge which mixture of MOMENT with the statistical models is tendentially better.

- Though MOMENT-Stat performs slightly better than MOMENT as a standalone, MOMENT with CAE is never compared with MOMENT-Stat with CAE. One could imagine that datasets dominated by low frequency anomaly signal would tendentially also rely on high level statistics, so the enhancement of the TFM might not provide any significant advantage when ensembled.

- The paper is lacking some focus. The CAE seems to be the central topic, but at the same time the statistical enhancement is introduced and near the end there is a quick study of forecasting TFMs. We feel it would have been better to have focused on CAE and try to support it more thoroughly, e.g. provide cleaner and more thorough definitions, add more details, like the relation of alpha with w, also study simple ensembles and properly compare the to CAE, further study how CAE is behaving on different datasets and picking its weighting.

**Questions:**

- (More a remark) Section 3.1 is quite messy. A lot of symbols and operations are used without being introduced and details are missing. It would be great to revise and improve it. The usage of intervals and their complements to filter series is nicely used though.

---

> ### Author Response · Authors · 2025-11-21
>
> ## General response
>
> We thank the Reviewer for the careful review and constructive suggestions, as well as the positive remarks on readability and clarity (S1), the use of realistic datasets and threshold-free metrics (S2), and the broad coverage of TFMs for anomaly detection (S3). **In the revised version, we unified the terminology, clarified the proofs, and conducted additional experiments, including multivariate settings and F1-score evaluation.**
>
> &nbsp;
>
> ## W1. CAE vs simple ensembling
>
> We propose to mitigate overgeneralization by ensembling a TFM with a
> statistical model and determining the ensemble weight based on the
> complexity of each instance. We agree with the reviewer that even a
> fixed-weight ensemble between a TFM and a statistical model can already
> be effective. Additionally, our goal is to show that CAE
> (Complexity-Aware Ensemble) provides a theoretically grounded and
> empirically stronger weighting rule.
>
> First, Sections 3 and Appendix A now
> explicitly connect the raw complexity gap
> $\Delta = L_{\mathrm{imp}} - L_{\mathrm{rec}}$ to high frequency energy
> and to anomaly gap growth. Theorem 3.1 proves that
> $\Delta \approx \sum_k \phi(k)\,b_k$ with coefficients $b_k$ that
> increase with frequency, and Theorem 3.2 links larger high frequency
> share to larger anomaly score separation. Theorem 3.1 and Theorem 3.2
> have been rewritten more clearly in the revision.
>
> Second, on the
> univariate TSB-AD-U benchmark, CAE improves over the best fixed weight
> $\alpha = 0.5$ for all five statistical backbones. For Sub-PCA the
> VUS-PR rises from $0.4557$ ($\alpha = 0.5$) to $0.4679$ with CAE, which
> also obtains the best overall rank among 25 baselines. A paired t-test
> over the 350 series yields $p \approx 0.07$, indicating that CAE more
> often selects useful weights than any single constant $\alpha$.
>
> Finally,
> to directly compare CAE with a simple fixed ensemble in the multivariate
> setting, we added the following table that reports naive ensembles with
> $\alpha = 0.5$ and CAE for each statistical backbone that supprots multivariate data:
>
> **Table 1. Comparison between naive ensemble and CAE on six multivariate datasets.**
>
> | **Detector** | **VUS-PR ($\alpha=0.5$)** | **VUS-PR CAE** | **VUS-ROC ($\alpha=0.5$)** | **VUS-ROC CAE** |
> |-------------|----------------------------------|----------------|-----------------------------------|-----------------|
> | HBOS        | 0.1751                           | **0.2535**     | 0.7099                            | **0.7324**      |
> | IForest     | 0.2553                           | **0.2996**     | **0.8070**                        | 0.7981          |
> | LOF         | 0.1091                           | **0.2122**     | 0.6389                            | **0.6796**      |
> | PCA         | 0.3132                           | **0.3433**     | **0.7834**                        | 0.7730          |
>
> As shown in Table 1, CAE consistently improves VUS-PR over the naive ensemble and often also improves VUS-ROC, especially for HBOS and LOF where the gains are substantial. This supports the importance of adaptively adjusting the ensemble weight based on complexity rather than using a single fixed value across datasets.
>
> &nbsp;
>
> ## W2. Relation between $\alpha$ and the ensemble weight $w$
>
> We clarify the relation between $\alpha$ and the ensemble weight in the revision.
> First, we define the raw complexity
> $$w(x) = \frac{L_{\mathrm{imp}}(x) - L_{\mathrm{rec}}(x)}{\lVert x \rVert_2^2},$$
> and obtain the normalized complexity
> $$\alpha(x) = \mathrm{QuantileTransform}\bigl(w(x)\bigr) \in [0,1].$$
> CAE then uses $\alpha(x)$ directly as the ensemble weight:
> $s_{\mathrm{CAE}}(x)  = \alpha(x)s_{\mathrm{TFM}}(x)  + \bigl(1 - \alpha(x)\bigr)s_{\mathrm{stat}}(x).$
>
> &nbsp;
>
> ## W3. MOMENT-Stat vs MOMENT with CAE
>
> All CAE results in the paper (e.g., Table 3 and Figure 1) already use MOMENT-Stat as the TFM branch, which we now state explicitly at the beginning of Section 4.5. This is important because Lemma 3.3 shows that TFMs with instance normalization are invariant to affine changes in mean and variance and thus cannot detect purely statistical anomalies. Augmenting the decoder input with $(\mu,\sigma)$ in MOMENT-Stat restores this sensitivity without retraining the encoder and improves the standalone VUS-PR from $0.3857$ for MOMENT (FT) to $0.3913$, moving from rank 6 to rank 3 among 25 baselines.
>
> &nbsp;
>
> ## W4. Focus and scope
>
> We thank the reviewer for this comment. Our intention was to show a broad range of TFM-related studies, but we agree that the main focus should be on CAE and will keep this clearer in the revised version.
>
> &nbsp;
>
> ## Q1. Section 3.1 clarity and notation
>
> We revised Section 3.1 to clarify the notation and definitions of the losses and the complexity metric, and moved technical details and assumptions to Appendix A so that the main text is easier to follow.

---

> > ### Comment · Reviewer_BRgs · 2025-11-26
> >
> > Thank you for responding to my questions. I have also adjusted the rating.

---

> > > ### Author Response · Authors · 2025-11-27
> > >
> > > Thank you for your time reviewing our paper and for adjusting the rating. We appreciate your constructive feedback.

---

### Author Response · Authors · 2025-11-30

**Dear Area Chair,**

We thank the reviewers for their constructive feedback. Based on the reviews, we have made significant improvements to the manuscript.

In brief, our study analyzes the overgeneralization and overstationarization issues that arise when applying time-series foundation models (TFMs) to anomaly detection, and proposes two remedies: a Complexity-Aware Ensemble (CAE) and Instance-Statistics Augmentation. Our model establishes SOTA performance among 26 baselines on the  [TSB-AD](https://github.com/TheDatumOrg/TSB-AD) leaderboard. We achieve this by introducing a theoretically grounded complexity metric $\alpha$ to adaptively adjust ensemble weights, and analytically demonstrating that instance normalization induces overstationarization, providing a justification for our augmentation strategy.

---

## 1. Strengths Highlighted by Reviewers

| Category          | Key Strength    | Reviewers        |
|--------------------|-------------------------------------------------------------------------------------------------------------------------------------------------------------------|------------------|
| Motivation         | Identifies the overgeneralization issue in TFMs and proposes a novel CAE guided by the metric $\alpha$.                                                          | aW8c, q2N4, zNh6 |
| Experiments        | Extensive evaluation on 23 univariate datasets using threshold-free metrics (VUS-PR / VUS-ROC). | aW8c, BRgs, q2N4 |
| Scope & Timeliness | One of the first systematic investigations of TFMs for anomaly detection, moving beyond isolated case studies.                                                   | BRgs, zNh6       |

---

## 2. Resolution of Reviewer Concerns

We summarize how our revisions specifically address the key concerns raised by the reviewers.

| Key Concern     | Resolution (Rebuttal Actions)     | Reviewers        |
|-------------------------------------------|-------------------------------------------------------------------------------------------------------------------------------------------------------------------------------------------------------------------------------------------------------------------|------------------|
| Clarity of Method Section                 | **Resolved:** Extensively revised Section 3.1 to clarify notations. We also addressed confusion regarding the ensemble weight  by formally defining the mapping from complexity metric via a quantile transform.       | BRgs, q2N4, zNh6 |
| Interpretation of Complexity ($\alpha$)   | **Resolved:** Theorem 3.1 interprets $\alpha$ as data complexity dominated by high-frequency components, while Theorem 3.2 proves that this higher complexity accelerates the separation between normal and anomalous scores.                                     | aW8c    |
| Experiments were limited to univariate.   | **Resolved:** Evaluated CAE on six multivariate datasets from TSB-AD-M. Our CAE consistently outperforms the naive ensemble model in terms of VUS-PR.                                                                                                                | aW8c, zNh6       |
| Unclear if CAE outperforms fixed weights ensemble. | **Resolved:** Paired t-tests ($p = 0.06$) on 350 univariate series and new multivariate results show that CAE outperforms fixed weights ensemble. Theorems 3.1 and 3.2 provide a theoretical foundation for the effectiveness of $\alpha$.     | BRgs   |
| Show benefits over ensembles of purely statistical methods. | **Resolved:** Compared CAE against the best purely statistical ensemble (Sub-PCA + SR). CAE consistently achieves higher VUS-PR (Appendix G).                                                                                                                    | zNh6             |
| Comparison with task-specific detectors and Best F1. | **Resolved:** On the TSB-AD leaderboard including Anomaly Transformer (VUS-PR 0.1195), our CAE (VUS-PR 0.4679) ranks first. We also report Best F1 (CAE 0.4330 vs. SOTA 0.42).                                                                                      | q2N4       |
| Sensitivity Analysis     | **Resolved:** Sensitivity analyses (Tables 3 & 4) confirm that our quantile-based weighting is robust to hyperparameter variations and outperforms entropy-based schemes.      | aW8c, zNh6       |
| Normalization & Mechanism  | **Resolved:** Added Lemma 3.3, formally proving that TFMs with instance normalization are invariant to affine shifts. This theoretically justifies our Instance Statistics Augmentation, which empirically improves MOMENT's standalone VUS-PR and rank.        | aW8c, q2N4        |

We appreciate your time and consideration regarding our rebuttal. By bridging theoretical analysis (Theorems 3.1 & 3.2 and Lemma 3.3) with extensive empirical validation, we hope this study will serve as a meaningful guideline for future research on adapting TFMs to anomaly detection.

---

### Meta-Review · Area_Chair_qRF9 · 2026-01-02

**Summary:**

The paper examines the use of time-series foundation models (TFMs) for anomaly detection, arguing that overgeneralization and overstationarization degrade reconstruction-based detectors. It proposes a complexity- and statistics-guided framework that combines TFMs with statistical detectors and reintroduces instance-level statistics lost through normalization.

Initial reviews raised concerns about whether the approach meaningfully improves over standard ensembling, the unclear definition of the complexity metric α, limited experimental support for some claims, a narrow evaluation scope, and issues with clarity and notation. In the rebuttal, the authors addressed most of these points, with multiple reviewers explicitly noting that their main concerns had been resolved.
The reviewer zNh6, who gave the lowest score, did not respond after the rebuttal, which had addressed most of their critiques. The primary unresolved issue is the study's limited scope, as the paper remains focused on reconstruction-based TFMs and does not extend the framework to other anomaly detection paradigms or broader TFM applications. While this is a valid limitation, it concerns scope and positioning rather than technical soundness or experimental validity.

Overall, the reviewers’ post-rebuttal inclination is borderline positive. Despite remaining scope limitations, the work constitutes a solid and timely contribution to the study of anomaly detection with time-series foundation models.

**Reviewer Concerns:**

BRgs: CAE advantage over simple ensembling is not clearly demonstrated. Addressed.

BRgs, aW8c: Definition of the complexity metric α, its properties, and its relation to the ensemble weight were unclear. Addressed.

BRgs: Lack of focus and methodological clarity. Addressed.

BRgs, zNh6, aW8c, q2N4: Some claims are insufficiently supported by experiments. Addressed.

zNh6, aW8c: Limited evaluation scope. Partly addressed.

q2N4: Presentation issues. Addressed.

**Reviewer Scores:**

BRgs: Probably would have raised from 4 to 6.

zNh6: Probably would have raised from 2 to 4.

aW8c: Probably would have stayed at 6.

q2N4: Probably would have raised from 4 to 6.

---

### Decision · Program_Chairs · 2026-01-26

Accept (Poster)